# OPTAMI: GLOBAL SUPERLINEAR CONVERGENCE OF HIGH-ORDER METHODS

**Dmitry Kamzolov**[1,2][*] **Artem Agafonov**[1,3]**, Dmitry Pasechnyuk**[1]**,**
**Alexander Gasnikov**[4,3,5]**, Martin Takáč**[1]

[1] Mohamed bin Zayed University of Artificial Intelligence, Abu Dhabi, UAE
[2] Toulouse School of Economics, Toulouse, France
[3] Moscow Institute of Physics and Technology, Dolgoprudny, Russia
[4] Innopolis University AI Research Center, Kazan, Russia
[5] Institute for System Programming RAS, Moscow, Russia

## ABSTRACT

Second-order methods for convex optimization outperform first-order methods in terms of theoretical iteration convergence, achieving rates up to $O(k^{-5})$ for highly-smooth functions. However, their practical performance and applications are limited due to their multi-level structure and implementation complexity. In this paper, we present new results on high-order optimization methods, supported by their practical performance. First, we show that the basic high-order methods, such as the Cubic Regularized Newton Method, exhibit global superlinear convergence for $\mu$-strongly star-convex functions, a class that includes $\mu$-strongly convex functions and some non-convex functions. Theoretical convergence results are both inspired and supported by the practical performance of these methods. Secondly, we propose a practical version of the Nesterov Accelerated Tensor method, called NATA. It significantly outperforms the classical variant and other high-order acceleration techniques in practice. The convergence of NATA is also supported by theoretical results. Finally, we introduce an open-source computational library for high-order methods, called OPTAMI. This library includes various methods, acceleration techniques, and subproblem solvers, all implemented as PyTorch optimizers, thereby facilitating the practical application of high-order methods to a wide range of optimization problems. We hope this library will simplify research and practical comparison of methods beyond first-order.

## 1 INTRODUCTION

In this paper, we consider the following unconstrained optimization problem:

$$\min_{x \in \mathbb{E}} f(x),  \tag{1}$$

where $\mathbb{E}$ is a $d$-dimensional real value space and $f(x)$ is a highly-smooth function

**Definition 1.1** *Function $f$ has $L_p$ - Lipschitz-continuous p-th derivative, if*

$$\|D^p f(x) - D^p f(y)\|_{op} \le L_p \|x - y\| \qquad \forall x, y \in \mathbb{E},  \tag{2}$$

*where $D^p f(x)$ is a p-th order derivative, and $\| \cdot \|_{op}$ is an operator norm.*

In the paper, we primarily focus on three main cases: $p = \{1; 2; 3\}$. We assume that the function $f$ is convex, although for some results, we relax this assumption to star-convexity. By $x^*$ we denote the minimum of $f$.

Second-order methods are widely used in optimization, finding applications in diverse fields such as machine learning, statistics, control, and economics (Polyak, 1987; Boyd & Vandenberghe, 2004; Nocedal & Wright, 1999; Nesterov, 2018). Historically, much of the research on second-order methods has focused on their local quadratic convergence. A well-known method achieving this rapid local rate is the classical Newton method (Newton, 1687; Raphson, 1697; Kantorovich, 1948b). However, it can diverge if the starting point is far from the solution (Nesterov, 1983, Example 1.2.3). To address this divergence issue, the Damped Newton method introduces a step-size (damping

---

[*]Corresponding author. e-mail: kamzolov.opt@gmail.com

coefficient) to ensure global convergence. However, the best-known global rate for the Damped Newton method is $O(T^{-1/3})$ (Berahas et al., 2022), which is slower than the gradient method's convergence $O(T^{-1})$. The Cubic Regularized Newton (CRN) method, introduced by Nesterov & Polyak (2006), was the first second-order method with a proper global convergence rate $O(T^{-2})$, outperforming the gradient method. Additionally, for strongly convex functions, it retains a quadratic local convergence rate, similar to the Newton method (Doikov & Nesterov, 2022). The introduction of CRN represented a significant milestone in the advancement of second-order optimization methods.

**Hessian approximations.** In large-scale optimization problems, computing the (inverse) Hessian or solving a linear system can be computationally expensive. Thus, it is natural to consider inexact or stochastic algorithms to reduce these overheads. In convex optimization, several studies have explored globally convergent second-order methods with inexact Hessians (Ghadimi et al., 2017), higher-order methods with inexact and stochastic derivatives (Agafonov et al., 2024a;b), and adaptive stochastic methods (Antonakopoulos et al., 2022). Recently, Quasi-Newton (QN) Hessian approximations have been integrated into global second-order methods, resulting in algorithms that outperform first-order methods — even when relying solely on first-order information (Kamzolov et al., 2023b; Jiang et al., 2023; Scieur, 2023; Jiang et al., 2024). Furthermore, numerous second-order approximation techniques have been developed for training neural networks, often surpassing state-of-the-art first-order methods. Notable examples include Shampoo (Gupta et al., 2018), SOAP (Vyas et al., 2025), and SOPHIA (Liu et al., 2024), which showcase the effectiveness of second-order approaches in practical applications and benchmarks[1] (Dahl et al., 2023). Such potential motivates us to study second-order methods.

**Accelerations.** The Cubic Regularized Newton is the basic method in the line-up of second-order methods. There are two main directions for its improvement: accelerated second-order methods, including Nesterov-type acceleration (Nesterov, 2008; 2021b), near-optimal acceleration Monteiro & Svaiter (2013); Gasnikov et al. (2019b), and optimal acceleration Kovalev & Gasnikov (2022); Carmon et al. (2022); and third-order methods with superfast subsolver, which allows making a third-order step without computation of third-order derivative (Nesterov, 2021b;c;a; Kamzolov, 2020).

## 1.1 OPTAMI: PRACTICAL PERFORMANCE OF HIGH-ORDER METHODS

The theoretical results mentioned above highlight the significant potential of second-order methods in optimization. However, their practical adoption remains limited due to the computational cost of calculating second derivatives, the variety of acceleration techniques, and the use of different Hessian approximation methods to reduce iteration costs. To address these challenges, we introduce OPTAMI[2], a unified library implemented in PyTorch for second-order and higher-order optimization methods.

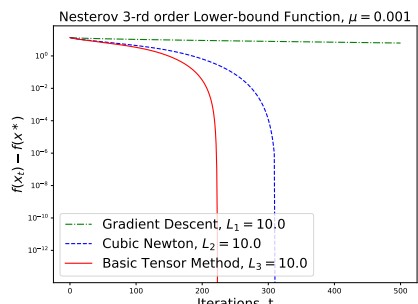

One particular goal of this library is a direct *comparison of a wide variety of acceleration techniques*, which include Nesterov acceleration (Nesterov, 2021b) with a rate $O(T^{-(p+1)})$; Near-Optimal Monteiro-Svaiter Acceleration (Monteiro & Svaiter, 2013; Bubeck et al., 2019; Gasnikov et al., 2019b; Kamzolov, 2020) with a rate $\tilde{O}(T^{-(3p+1)/2})$; Near-Optimal Proximal-Point Acceler-

Figure 1: Third-order Nesterov's lower-bound function. Cubic Newton and Basic Tensor method converge *superlinearly*. In contrast, GD demonstrates *linear* rate.

ation (Nesterov, 2021a) with the rate $\tilde{O}(T^{-(3p+1)/2})$; Optimal Acceleration (Kovalev & Gasnikov, 2022; Carmon et al., 2022) with a rate $O(T^{-(3p+1)/2})$ and more Nesterov (2023). Despite the theoretical advancements in these methods, the literature lacks a comprehensive practical comparison, especially for higher-order methods with $p = 3$.
In the process of developing the library, we encountered several open challenges.

**Methods exceed linear convergence in practice.** We observed in experiments that second-order and third-order methods often achieve superlinear convergence rates for $\mu$-strongly convex functions (Figure 1). From a theoretical standpoint, this is surprising. The lower bound is

---

[1]https://mlcommons.org/benchmarks/algorithms/
[2]https://github.com/OPTAMI/OPTAMI

$\Omega\left(\left(\frac{L_2 D}{\mu}\right)^{2/7} + \log\log\left(\frac{\mu^3}{L_2^2\varepsilon}\right)\right)$ for $\varepsilon \leq c_1\frac{\mu^3}{L_2^2} = c_1 r$ as established by Arjevani et al. (2019), where $r$ is the radius of quadratic convergence $\left\{x \in \mathbb{E} : f(x) - f^* \leq c_2 r = c_2\frac{\mu^3}{L_2^2}\right\}$ and $c_1, c_2$ are universal constants. The power $2/7$ corresponds to the optimal accelerated method. However, this lower bound applies only when $\varepsilon \leq c_1 r$, which corresponds to small values of $\varepsilon$. In the case when $\varepsilon > c_1 r$, meaning the desired accuracy exceeds the radius of the quadratic convergence region, it may be possible to achieve faster global rates of Cubic Regularized Newton method than linear convergence.

**Practical performance of accelerated methods.** We also observed that the Nesterov Accelerated Tensor Method (Nesterov, 2021b) performs worse or on par with its non-accelerated counterpart in practice. This contrasts with first-order methods, where acceleration is typically beneficial. These practical limitations lead to the method being underutilized (Scieur, 2023; Carmon et al., 2022; Antonakopoulos et al., 2022).

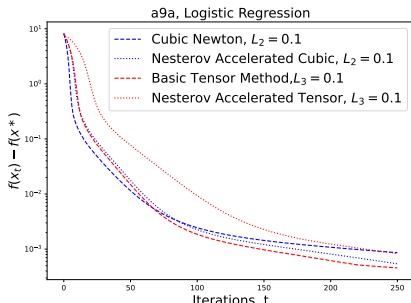

Figure 2: Comparison of Basic Methods vs Nesterov Accelerated Methods

In our work, alongside introducing the OPTAMI library, we aim to address these open challenges from both theoretical and practical perspectives.

**Contributions.** We summarize our key contributions as follows:

1. **Global Superlinear Convergence of Second and High-order methods.** Our main contribution is providing theoretical guarantees for *global superlinear convergence* of the Cubic Regularized Newton Method and the Basic Tensor Methods for $\mu$-strongly star-convex functions. These theoretical results are validated by practical performance. These results are a significant improvement over the current state-of-the-art in second-order methods.

2. **Nesterov Accelerated Tensor Method with $A_t$-Adaptation (NATA).** We propose a new practical variant of the Nesterov Accelerated Tensor Method, called NATA. This method addresses the practical limitations of the classical version of acceleration for high-order methods. We demonstrate the superior performance of NATA compared to both the classical Nesterov Accelerated Tensor Method and Basic Tensor Method for $p = 2$ and $p = 3$. We also prove a convergence theorem for NATA that matches the classical convergence rates.

3. **Comparative Analysis of High-Order Acceleration Methods.** We provide a practical comparison of state-of-the-art (SOTA) acceleration techniques for high-order methods, with a focus on the cases $p = 2$ and $p = 3$. Our experiments show that the proposed NATA method consistently outperforms all SOTA acceleration techniques, including both optimal and near-optimal methods.

4. **Open-Source Computational Library for Optimization Methods (OPTAMI).** We introduce *OPTAMI*, an open-source library for high-order optimization methods. It facilitates both practical research and applications in this field. Its modular architecture supports various combinations of acceleration techniques with basic methods and their subsolvers. All methods are implemented as PyTorch optimizers. This allows for seamless application of high-order methods to a wide range of optimization problems, including neural networks.

## 2 METHODS AND NOTATION

**Notation.** In the paper, we consider a $d$-dimensional real value space $\mathbb{E}$. $\mathbb{E}^*$ is a dual space, composed of all linear functionals on $\mathbb{E}$. For a functional $g \in \mathbb{E}^*$, we denote by $\langle g, x \rangle$ its value at $x \in \mathbb{E}$. For $p \geq 1$, we define $D^p f(x)[h_1, \ldots, h_p]$ as a directional $p$-th order derivative of $f$ along $h_i \in \mathbb{E}, i = 1, \ldots, p$. If all $h_i = h$ we simplify $D^p f(x)[h_1, \ldots, h_p]$ as $D^p f(x)[h]^p$. So, for example, $D^1 f(x)[h] = \langle \nabla f(x), h \rangle$ and $D^2 f(x)[h]^2 = \langle \nabla^2 f(x)h, h \rangle$. Note, that $\nabla f(x) \in \mathbb{E}^*$, $\nabla^2 f(x)h \in \mathbb{E}^*$. Now, we introduce different norms for spaces $\mathbb{E}$ and $\mathbb{E}^*$. For a self-adjoint positive-definite operator $B : \mathbb{E} \to \mathbb{E}^*$, we can endow these spaces with conjugate Euclidian norms:

$$\|x\| = \langle Bx, x \rangle^{1/2}, \quad x \in \mathbb{E}, \qquad \|g\|_* = \langle g, B^{-1}g \rangle^{1/2}, \quad g \in \mathbb{E}^*.$$

So, for an identity matrix $B = I$, we get the classical 2-norm $\|x\|_2 = \|x\|_I = \langle x, x \rangle^{1/2}$. We denote $\mathbf{e} \in \mathbb{R}^d$ as a vector of all ones and $\mathbf{0} \in \mathbb{R}^d$ as a vector of all zeroes.

We introduce two types of distance measures between the starting point and the solution: for non-accelerated methods, we consider the diameter of the level set $\mathcal{L} = \{x \in \mathbb{E} : f(x) \leq f(x_0)\}$

$$D = \max_{x \in \mathcal{L}} \|x - x^*\|; \tag{3}$$

and for accelerated methods, we use the Euclidean distance given by

$$R = \|x_0 - x^*\|. \tag{4}$$

## 2.1 Methods in OPTAMI library

In this subsection, we present a detailed overview of the core methods implemented in the OPTAMI library. Second-order methods have a more complicated structure. The library's design is structured into three hierarchical levels: basic methods, subsolvers, and accelerations. This modular architecture ensures flexibility, extensibility, and adaptability to a variety of optimization tasks. It allows users to combine multiple basic methods with various accelerations and subsolvers without the need to implement entire methods from scratch. We leave technical details of the subsolvers to Appendix C.

BASIC METHODS. The Basic methods are the foundational building blocks of the library. These monotone, non-accelerated methods form the backbone for constructing more sophisticated accelerated algorithms. Below, we outline the primary basic methods available in the library.
**Newton method.** The classical (Damped) Newton method is defined as follows:

$$x_{t+1} = x_t - \gamma_t \left[\nabla^2 f(x_t)\right]^{-1} \nabla f(x_t), \tag{5}$$

where $\gamma_t \in \mathbb{R}_+$ is a step-size or damping coefficient. The Newton step originates from the second-order Taylor expansion $\Phi_2(x, x_t)$:

$$x_{t+1} = \underset{x \in \mathbb{E}}{\operatorname{argmin}} \left\{\Phi_2(x, x_t) = f(x_t) + \langle \nabla f(x_t), x - x_t \rangle + \langle \nabla^2 f(x_t)(x - x_t), x - x_t \rangle\right\}. \tag{6}$$

The solution of this problem corresponds to (5) with $\gamma_t = 1$. The Newton method lacks global convergence, while the Damped Newton method exhibits a slow global convergence rate of $O(T^{-1/3})$. This is because the approximation $\Phi_2(x, x_t)$ is not guaranteed to be an upper bound for $f$, meaning it is possible that $f(x) > \Phi_2(x, x_t)$.
**Cubic Regularized Newton method.** To address this issue, the Cubic Regularized Newton (CRN) method was proposed

$$x_{t+1} = \operatorname{argmin}_{y \in \mathbb{E}} \left\{\Omega_{M_2}(x, x_t) = \Phi_2(x, x_t) + \frac{M_2}{6}\|x - x_t\|^3\right\}. \tag{7}$$

For the function $f(x)$ with $L_2$-Lipschitz Hessian, the model $\Omega_{M_2}(y, x_t)$ is an upper bound of the function $f(x)$ for $M_2 \geq L_2$; hence $\Omega_{M_2}(x, x_t) \geq f(x)$. This method is the first second-order method with a global convergence rate of $O\left(\frac{M_2 D^3}{T^2}\right)$, which is faster than the Gradient Method (GM).
**Basic Tensor method.** High-order Taylor approximation of a function $f$ can be written as follows:

$$\Phi_{x,p}(y) = f(x) + \sum_{k=1}^p \frac{1}{k!} D^k f(x)[y-x]^k, \quad x, y \in \mathbb{E}, \tag{8}$$

where, for $p = 1$, we simplify notation to $\Phi_x(y)$. From (2), we can get the next upper-bound of the function $f(x)$ (Nesterov, 2018; 2021b)

$$|f(y) - \Phi_{x,p}(y)| \leq \frac{L_p}{(p+1)!}\|y - x\|^{p+1}, \tag{9}$$

which leads us to the high-order model

$$\Omega_{x,M_p}(y) = \Phi_{x,p}(y) + \frac{M_p}{(p+1)!}\|y - x\|^{p+1}. \tag{10}$$

Now, we can formulate the Basic Tensor method

$$x_{t+1} = \operatorname{argmin}_{y \in \mathbb{E}} \left\{\Omega_{x_t,M_p}(y)\right\}, \tag{11}$$

where $M_p \geq pL_p$. For $p = 1$ and $M_1 \geq L_1$, it is the gradient descent step $x_{t+1} = x_t - \frac{1}{M_1}\nabla f(x_t)$ with the convergence rate $O\left(\frac{M_1 R^2}{T}\right)$ for convex functions. For $p = 2$ and $M_2 \geq L_2$, it is a CRN Method from (7). For $p = 3$ and $M_3 \geq 3L_3$, it is a Basic Third-order Method (Nesterov, 2021b):

$$x_{t+1} = x_t + \underset{h \in \mathbb{E}}{\operatorname{argmin}} \left\{f(x_t) + \nabla f(x_t)[h] + \frac{1}{2}\nabla^2 f(x_t)[h]^2 + \frac{1}{6}D^3 f(x_t)[h]^3 + \frac{M_3}{24}\|h\|^4\right\}, \tag{12}$$

with the convergence rate $O\left(\frac{M_3 D^4}{T^3}\right)$. The step (12) can be performed with almost the same computational complexity (up to a logarithmic factor) by using the Bregman Distance Gradient Method as a subsolver (Nesterov, 2021b;c). The details are written in the Appendix C.1.

ACCELERATIONS. Compared to first-order methods, second-order and higher-order methods achieve three types of acceleration rates: Nestrov-type acceleration with the rate $O\left(T^{-(p+1)}\right)$, nearly-optimal acceleration $\tilde{O}\left(T^{-(3p+1)/2}\right)$, and optimal one $O\left(T^{-(3p+1)/2}\right)$, where $\tilde{O}(\cdot)$ means up to a logarithmic factor. OPTAMI library includes four key variants of acceleration techniques:

- Nesterov Accelerated Tensor Method (Algorithm 1) with a rate $O(T^{-(p+1)})$ (Nesterov, 2021b);
- Near-Optimal Tensor Acceleration (Algorithm 5) with a rate $\tilde{O}(T^{-(3p+1)/2})$ (Bubeck et al., 2019; Gasnikov et al., 2019b; Kamzolov, 2020);
- Near-Optimal Proximal-Point Acceleration Method with Segment Search (Algorithm 6) with the rate $\tilde{O}(T^{-(3p+1)/2})$ (Nesterov, 2021a);
- Optimal Acceleration (Algorithm 7) with a rate $O(T^{-(3p+1)/2})$ (Kovalev & Gasnikov, 2022).

These methods are presented in detail in Section 3.1 for Nesterov acceleration, and in Appendix D for the remaining algorithms.

## 3 IMPROVING PRACTICAL PERFORMANCE OF ACCELERATED METHODS

While accelerated second-order and higher-order methods provide provable theoretical advancements over their non-accelerated counterparts, a detailed comparison of their practical performance seems to be underexplored in the literature. Notably, techniques like Nesterov acceleration, which are highly effective for first-order methods, can slow down second-order and higher-order methods, particularly in the initial stages (Scieur, 2023; Carmon et al., 2022; Antonakopoulos et al., 2022). To illustrate this, we present a practical example using the logistic regression problem (Figure 2). The accelerated versions appear slower, which contradicts the theoretical expectations.

In this section, we first introduce a novel algorithm, NATA, that enhances the practical performance of the Nesterov Accelerated Tensor Method while maintaining the same theoretical guarantees. We then provide a comprehensive computational comparison of five different acceleration techniques for second-order and higher-order optimization.

### 3.1 NESTEROV ACCELERATED TENSOR METHOD WITH $A_t$-ADAPTATION (NATA)

---

**Algorithm 1** Nesterov Accelerated Tensor Method

1: **Input:** $x_0 = v_0$ is starting point, constant $M_p$, $\psi_0(z) = \frac{1}{p+1}\|z - x_0\|^{p+1}$, total number of iterations $T$, and sequence $A_t$.
2: **for** $t \geq 0$ **do**
3:      $a_{t+1} = A_{t+1} - A_t$
4:      $y_t = \frac{A_t}{A_{t+1}}x_t + \frac{a_{t+1}}{A_{t+1}}v_t$
5:      $x_{t+1} = \operatorname{argmin}_{y \in \mathbb{E}}\left\{\Omega_{y_t, M_p}(y)\right\}$
6:      $\psi_{t+1}(z) = \psi_t(z) + a_{t+1}[f(x_{t+1}) + \langle\nabla f(x_{t+1}), z - x_{t+1}\rangle]$
7:      $v_{t+1} = \operatorname{argmin}_{x \in \mathbb{E}}\psi_{t+1}(z)$
8: **end for**
9: **return** $x_{T+1}$

---

In this subsection, we investigate the causes of the under-performance of Nesterov Accelerated Tensor method and propose a solution. We begin by revisiting Algorithm 1, with further details provided in Appendix D.1. According to the theoretical convergence result $f(x_t) - f(x^*) \leq \frac{\|x^* - x_0\|^{p+1}}{(p+1)A_t}$ from (Nesterov, 2021c, Theorem 2.3), the sequence $A_t$ is directly connected with the method's performance - the larger the $A_t$, the faster the convergence. Therefore, our goal is to maximize $A_t$. Theoretically, $A_t$ should be defined as $A_t = \frac{\nu_p}{L_p}t^{p+1}$, where $\nu_2 = \frac{1}{24}$ for $M_2 = L_2$ and $\nu_3 = \frac{5}{3024}$ for $M_3 = 6L_3$. However, the values of $\nu_p$ appear to be quite small, which limits the speed of convergence. Can these values be increased? The

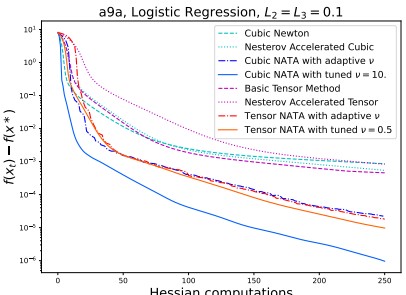

Figure 3: Basic and Nesterov Accelerated Methods vs new NATA Methods.

answer is yes. We propose the Nesterov Accelerated Tensor Method with $A_t$-Adaptation, which selects these parameters more aggressively, leading to faster convergence.

**Theorem 3.1** *For convex function $f$ with $L_p$-Lipschitz-continuous $p$-th derivative, to find $x_T$ such that $f(x_T) - f(x^*) \leq \varepsilon$, it suffices to perform no more than $T \geq 1$ iterations of the Nesterov Accelerated Tensor Method with $A_t$-Adaptation (NATA) with $M_p \geq pL_p$ (Algorithm 2), where*

$$T = O\left(\left(\frac{M_p R^{p+1}}{\varepsilon}\right)^{\frac{1}{p+1}} + \log_\theta\left(\frac{\nu^{\max}}{\nu^{\min}}\right)\right). \tag{13}$$

---

**Algorithm 2** Nesterov Accelerated Tensor Method with $A_t$-Adaptation (NATA)

---

1: **Input:** $x_0 = v_0$ is starting point, $\psi_0(z) = \frac{1}{p+1}\|z - x_0\|^{p+1}$, constant $M_p$, total number of
   iterations $T$, $\tilde{A}_0 = 0$, $\nu^{\min} = \nu_p$, $\nu^{\max} \geq \nu_p$ is a maximal value of $\nu$, $\theta > 1$ is a scaling
   parameter for $\nu$, and $\nu_0 \leq \nu^{\max}$ is a starting value of $\nu$.
2: **for** $t \geq 0$ **do**
3:    $\nu^t = \nu^t \theta$
4:    **repeat**
5:       $\nu^t = \max\left\{\frac{\nu^t}{\theta}, \nu^{\min}\right\}$
6:       $\tilde{a}_{t+1} = \frac{\nu^t}{M_p}((t+1)^{p+1} - t^{p+1})$ and $\tilde{A}_{t+1} = \tilde{A}_t + \tilde{a}_{t+1}$
7:       $y_t = \frac{\tilde{A}_t}{\tilde{A}_{t+1}}x_t + \frac{\tilde{a}_{t+1}}{\tilde{A}_{t+1}}v_t$
8:       $x_{t+1} = \operatorname{argmin}_{y \in \mathbb{E}}\left\{\Omega_{y_t, M_p}(y)\right\}$
9:       $\psi_{t+1}(z) = \psi_t(z) + \tilde{a}_{t+1}[f(x_{t+1}) + \langle \nabla f(x_{t+1}), z - x_{t+1}\rangle]$
10:      $v_{t+1} = \operatorname{argmin}_{z \in \mathbb{E}} \psi_{t+1}(z)$
11:    **until** $\psi_{t+1}(v_{t+1}) < \tilde{A}_{t+1}f(x_{t+1})$
12:    $\nu^{t+1} = \min\{\nu^t\theta, \nu^{\max}\}$
13: **end for**
14: **return** $x_{T+1}$

---

The proof is presented in the Appendix D.2. The established convergence rate of NATA matches the original method, with an additional factor of $\log_\theta\left(\frac{\nu^{\max}}{\nu^{\min}}\right)$ accounting for the adaptation of $\nu_t$. Next, we demonstrate the practical improvements of NATA compared to the classical methods. As shown in Figure 3, one can see that the Cubic and Tensor variants of NATA significantly outperform the classical Basic and Nesterov Accelerated Methods. We also included versions of Cubic and Tensor NATA with fixed $\nu^t = 10$ and $\nu^t = 0.5$, respectively, where $\nu^t$ is an additional tunable hyperparameter. This more aggressive variant of NATA can exhibit even faster practical performance, though it may diverge if $\nu^t$ is not chosen carefully.

## 3.2 COMPUTATIONAL COMPARISION OF ACCELERATION METHODS

We now present a practical comparison of various acceleration techniques for tensor methods in convex optimization, including Nesterov acceleration, near-optimal and optimal accelerations, as well as the newly proposed algorithm, NATA. Specifically, our experiments focus on logistic regression, defined as:

$$f(x) = \frac{1}{n}\sum_{i=1}^{n}\log\left(1 + e^{-b_i\langle a_i, x\rangle}\right) + \frac{\mu}{2}\|x\|_2^2, \tag{14}$$

where $a_i \in \mathbb{R}^d$ are data features and $b_i \in \{-1; 1\}$ are data labels for $i = 1, \ldots, n$. We evaluate performance on the a9a dataset in Figure 4 with regularizer $\mu = 0$ and $\mu = 10^{-4}$ in Figure 5.

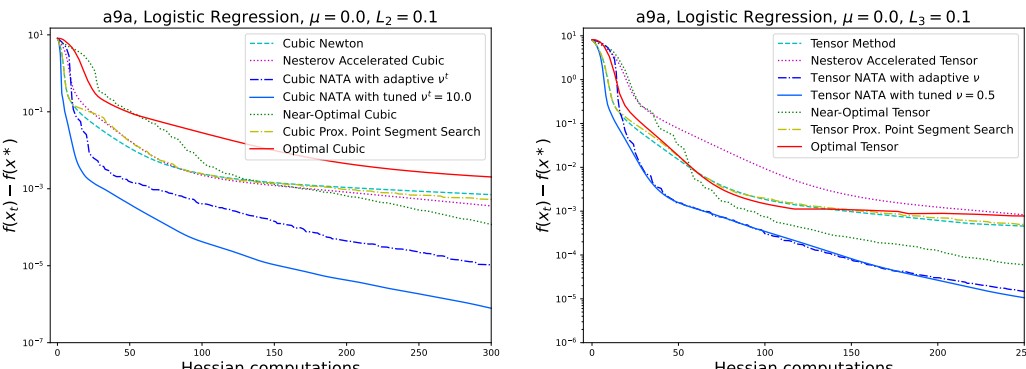

Figure 4: Comparison of different cubic and tensor acceleration methods on Logistic Regression for `a9a` dataset from the starting point $x_0 = 3\mathbf{e}$, where $\mathbf{e}$ is a vector of all ones.

Let us now discuss the performance of the methods. The new NATA acceleration outperforms all other methods. We attribute this to NATA's strategy of maximizing $\tilde{a}_t$ and $\tilde{A}_t$, which enables even faster convergence in the later stages. The second-best performer is the Near-Optimal Acceleration method. Although it struggles initially due to a large number of line-search iterations per step, it

gradually requires fewer line-search iterations — less than two per step on average — as parameters from previous line-search steps become well-suited for the current iteration. With fewer line-search iterations, the method accelerates and outpaces the remaining competitors. A promising direction for improving this method would be to refine the line-search process through an advanced line-search strategy. Next, the Nesterov Accelerated method starts off slower than the basic method without acceleration. Eventually, the method accelerates and overtakes the basic version but only for the Cubic version, as $\nu_3$ is too small for tensor methods. Near-Optimal Proximal-Point Acceleration Method with Segment Search performs very similarly to Basic Methods with only improvement in strongly convex case. It has much fewer iterations, but it does a safe segment search with an average of 3 Basic steps per search. Lastly, the Optimal Acceleration method performs the worst in practice. We believe the main issue lies in the internal parameters, which need tuning and adaptation, as we used the theoretical parameters in our implementation. This leads to many inner iterations without significant global progress. Improving these parameters presents an open question for future research. More details can be found in the Appendix E.

In Figure 5, both basic optimization methods and certain accelerated variants appear to exhibit *global superlinear convergence*, accelerating with each iteration even when far from the solution. This observation naturally raises an important question: Can we theoretically prove that second-order methods achieve *global superlinear convergence*? We address this question in the following section.

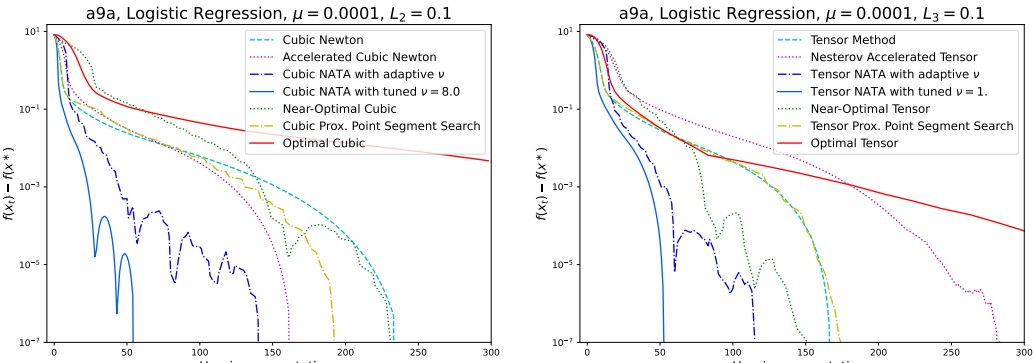

Figure 5: Comparison of different cubic and tensor acceleration methods on regularized Logistic Regression for `a9a` dataset and $\mu = 10^{-4}$ from the starting point $x_0 = 3\mathbf{e}$.

## 4    GLOBAL SUPERLINEAR CONVERGENCE OF HIGH-ORDER METHODS FOR STRONGLY STAR-CONVEX FUNCTIONS

In this section, we establish the global superlinear convergence of high-order methods for strongly star-convex functions. We begin by defining global superlinear convergence.

**Definition 4.1** *A method is said to exhibit a **global superlinear convergence rate** with respect to the functional gap if there exists a sequence $\zeta_t$ for all $t \in \{0, \ldots, T\}$ such that*

$$\frac{f(x_{t+1}) - f^*}{f(x_t) - f^*} \leq \zeta_t, \qquad 1 > \zeta_t > \zeta_{t+1} \quad \forall t \in \{0, \ldots, T\}, \quad and \quad \zeta_t \to 0 \ for \ t \to +\infty. \quad (15)$$

The essence of this definition lies in the fact that the scaling coefficient $\zeta_t$ decreases with each iteration. If $\zeta_t$ remains constant, the method achieves linear convergence. Conversely, if $\zeta_t$ increases over time (i.e., $\zeta_t < \zeta_{t+1}$), the convergence becomes sublinear. Additionally, we introduce the values $\alpha_t = 1 - \frac{f(x_{t+1}) - f^*}{f(x_t) - f^*} \leq 1$, which typically represent the per-iteration convergence rate from $f(x_{t+1}) - f^* \leq (1 - \alpha_t)(f(x_t) - f^*)$. The larger $\alpha_t$ means faster convergence. As for constant $\alpha \leq \alpha_t$, the method takes a total number of $T = O\left(\alpha^{-1} \log\left(\frac{f(x_0) - f^*}{\varepsilon}\right)\right)$ iterations to reach $\varepsilon$-solution, where $f(x_{T+1}) - f^* \leq \varepsilon$. For example, gradient descent exhibits global linear convergence for strongly convex functions with $\zeta_t = 1 - \alpha = 1 - \frac{\mu}{L_1 + \mu}$.

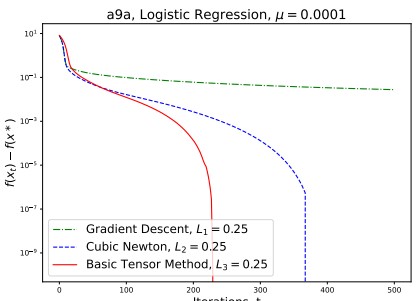

Figure 6: Cubic Newton and Basic Tensor method have areas of *superlinear* convergence. In contrast, GD demonstrates *linear* rate.

Now, to get some intuition on the performance of the methods, we begin with two simple and classical examples: the $l_2$-regularized logistic regression problem and the $l_2$-regularized Nesterov's

lower-bound function. The $l_2$-regularized third order Nesterov's lower-bound function from Nesterov (2021b) has the next form

$$f(x) = \frac{1}{4}\sum_{i=1}^{d-1}(x_i - x_{i+1})^4 - x_1 + \frac{\mu}{2}\|x\|_2^2. \tag{16}$$

Figures 1, 6 illustrate that both the Cubic Newton method and Basic Tensor method have areas of superlinear convergence where the graphics are going down faster with each iteration (concave downward). In contrast, gradient descent demonstrates linear convergence. To verify the behavior of these methods, we plot the values $\alpha_t = 1 - \frac{f(x_{t+1})-f^*}{f(x_t)-f^*} \leq 1$.

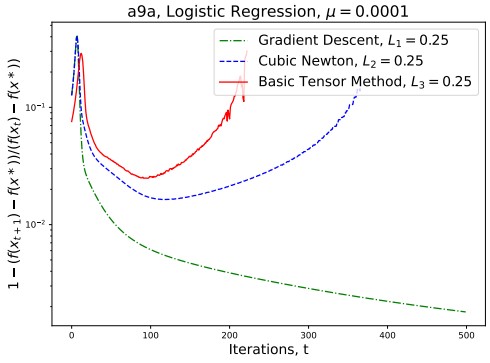
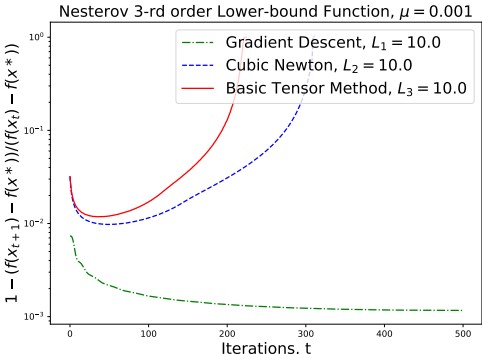

(a) Logistic Regression for a9a dataset starting from the point $x_0 = 3\mathbf{e}$ with $\mu = 10^{-4}$ regularizer.

(b) Third-order Nesterov's lower-bound function starting from the point $x_0 = \mathbf{0}$ with $\mu = 10^{-3}$ regularizer.

Figure 7: Comparison of the basic methods by the relative value $1 - \frac{f(x_{t+1})-f^*}{f(x_t)-f^*}$.

In Figure 7, we observe that at the beginning, all methods slow down for both cases. This phase corresponds to the region where the function's decrease guarantee for (star-)convex functions outperforms the function's decrease guarantee for strongly (star-)convex functions. For example, in the case of gradient descent, this occurs when the guarantee $f(x_{t+1}) \leq f(x_t) - \frac{1}{2L_1}\|\nabla f(x_{t+1})\|^2$ is better than $f(x_{t+1}) - f^* \leq \left(1 - \frac{\mu}{\mu+L_1}\right)(f(x_t) - f^*)$. Despite this region, gradient descent still has global linear convergence for strongly (star-)convex function. As iterations proceed, gradient descent stabilizes around $\alpha_t = 10^{-3}$, which corresponds to the theoretical convergence rate $\kappa$. The Cubic Newton method and the Basic Tensor method, however, start to accelerate and switch to a superlinear convergence rate. This practical performance gives the intuition for the global superlinear convergence of high-order methods.

Now, we present the theoretical results demonstrating that basic high-order methods indeed have a global superlinear convergence for $\mu$-strongly star-convex functions.

**Definition 4.2** *Let $x^*$ be a minimizer of the function $f$. For $q \geq 2$ and $\mu_q \geq 0$, the function $f$ is $\mu_q$-**uniformly star-convex of degree** $q$ with respect to $x^*$ if for all $x \in \mathbb{R}^d$ and $\forall \alpha \in [0,1]$*

$$f(\alpha x + (1-\alpha)x^*) \leq \alpha f(x) + (1-\alpha)f(x^*) - \frac{\alpha(1-\alpha)\mu_q}{q}\|x - x^*\|^q. \tag{17}$$

*If $q = 2$ then the function $f$ is $\mu$-**strongly star-convex** with respect to $x^*$. If $\mu_q = 0$ then the function $f$ is **star-convex** with respect to $x^*$. From this definition, we can additionally get the next useful inequality sometimes called q-order growth condition*

$$\frac{\mu_q}{q}\|x - x^*\|^q \leq f(x) - f(x^*). \tag{18}$$

We start with a simplified version of the theorem which includes the linear convergence and then we present the full version.

**Theorem 4.3** *For $\mu$-strongly star-convex (17) function $f$ with $L_2$-Lipschitz-continuous Hessian (2), Cubic Regularized Newton Method from (7) with $M_2 \geq L_2$ converges with the rate*

$$f(x_{t+1}) - f^* \leq (1 - \alpha_t)(f(x_t) - f^*), \tag{19}$$

*for all* $\quad \alpha_t \in [0; \alpha_t^*], \quad$ *where* $\quad \alpha_t^* = \frac{-1+\sqrt{1+4\kappa_t}}{2\kappa_t} \quad$ *and* $\quad \kappa_t = \frac{(M_2+L_2)\|x_t-x^*\|}{3\mu}. \tag{20}$

*This range includes the classical linear rate*

$$f(x_t) - f(x^*) \leq (1 - \alpha^{low})^t(f(x_0) - f(x^*)) \quad for \quad \alpha^{low} = \min\left\{\frac{1}{2}; \sqrt{\frac{3\mu}{4(M_2+L_2)D}}\right\} \tag{21}$$

*Proof.* We start the proof by using an upper-bound (9)

$$f(x_{t+1}) \overset{(9)}{\leq} \Phi_{x_t,2}(x_{t+1}) + \tfrac{L_2}{6}\|x_{t+1} - x_t\|^3 \overset{(7)}{\leq} \min_{y \in \mathbb{R}^n}\left\{\Phi_{x_t,2}(y) + \tfrac{M_2}{6}\|y - x_t\|^3\right\}$$

$$\overset{(9)}{\leq} \min_{y \in \mathbb{R}^n}\left\{f(y) + \tfrac{M_2+L_2}{6}\|y - x_t\|^3\right\} \overset{y=x_t+\alpha_t(x^*-x_t)}{\leq} f((1-\alpha_t)x_t + \alpha_t x^*) + \alpha_t^3\tfrac{M_2+L_2}{6}\|x^* - x_t\|^3$$

$$\overset{(17)}{\leq} (1-\alpha_t)f(x_t) + \alpha_t f(x^*) - \tfrac{\alpha_t(1-\alpha_t)\mu}{2}\|x_t - x^*\|^2 + \alpha_t^3\tfrac{M_2+L_2}{6}\|x_t - x^*\|^3.$$

From the second inequality, we get that the method is monotone and $f(x_{t+1}) \leq f(x_t)$. Now, by subbing $f(x^*)$ from both sides, we get

$$f(x_{t+1}) - f(x^*) \leq (1-\alpha_t)\left(f(x_t) - f(x^*)\right) - \tfrac{\alpha_t}{2}\|x_t - x^*\|^2\left((1-\alpha_t)\mu - \alpha_t^2\tfrac{M_2+L_2}{3}\|x_t - x^*\|\right),$$

By choosing $\alpha_t$ such that

$$\alpha_t^2\tfrac{M_2+L_2}{3}\|x_t - x^*\| + \mu\alpha_t - \mu \leq 0, \tag{22}$$

we get (19). By solving the quadratic inequality (22), we get that the method (7) converges with the rate (19) for all (20). Next, we present Lemma 4.4 with the useful properties of $\alpha_t^*$ from (20). The more general Lemma B.2 with the detailed proof is in Appendix B.

**Lemma 4.4** *For $z > 0$, the function*

$$\alpha^*(z) = \tfrac{-1+\sqrt{1+4z}}{2z} \tag{23}$$

*is bounded by the following lower and upper bounds*

$$\min\left\{1, \tfrac{1}{\sqrt{z}}\right\} > \alpha^*(z) > \min\left\{\tfrac{1}{2}; \tfrac{1}{2\sqrt{z}}\right\}, \tag{24}$$

*and it is monotonically decreasing*

$$\forall z, y > 0 : \quad z < y \quad \Rightarrow \quad \alpha^*(z) > \alpha^*(y). \tag{25}$$

The convergence rate is well-defined as $0 < \alpha_t^* \leq 1$ from (24). As $\|x_t - x^*\| \leq D$ from (3) and $\alpha^{low} \leq \alpha^*$ by (24), we get the linear convergence rate (21). □

Now, we move to the second theorem and prove the global superlinear convergence. The main idea of the proof is to observe that $\|x_t - x^*\|$ in (20) decreases for $\mu$-strongly star-convex functions. This property allows us to show that $\kappa_t$ is decreasing, and hence $\alpha_t^*$ is increasing from (25), leading to superlinear convergence.

**Theorem 4.5** *For $\mu$-strongly star-convex (17) function $f$ with $L_2$-Lipschitz-continuous Hessian (2), Cubic Regularized Newton Method from (7) with $M_2 \geq L_2$ converges globally superlinearly as defined in (15) with $\zeta_t = 1 - \alpha_t^{sl}$*

$$f(x_{t+1}) - f^* \leq (1 - \alpha_t^{sl})\left(f(x_t) - f^*\right), \tag{26}$$

*where*

$$\alpha_t^{sl} = \tfrac{-1+\sqrt{1+4\kappa_t^{sl}}}{2\kappa_t^{sl}} \quad for \quad \kappa_t^{sl} = \tfrac{(M_2+L_2)\sqrt{2}}{3\mu^{3/2}}(1 - \alpha^{low})^{t/2}\left(f(x_0) - f(x^*)\right)^{1/2}. \tag{27}$$

*The aggregated convergence rate for $T \geq 1$ equals to*

$$f(x_T) - f(x^*) \leq (f(x_0) - f(x^*))\prod_{t=1}^{T}(1 - \alpha_t^{sl}). \tag{28}$$

*Proof.* From $\mu$-strongly star-convexity (17), we can upper-bound $\|x_t - x^*\|$ in (20) by

$$\|x_t - x^*\| \leq \left(\tfrac{2}{\mu}\left(f(x_t) - f(x^*)\right)\right)^{1/2} \overset{(21)}{\leq} \left(\tfrac{2}{\mu}\left((1 - \alpha^{low})^t(f(x_0) - f(x^*))\right)\right)^{1/2}.$$

So, we got that $\|x_t - x^*\|$ is linearly decreasing to zero. From that, we get a new superlinear $\alpha_t^{sl} \leq \alpha_t^*$ from (27). As $\kappa_t^{sl}$ is getting smaller within each iteration $\kappa_t^{sl} > \kappa_{t+1}^{sl}$, we get that $\alpha(\kappa_t^{sl}) < \alpha(\kappa_{t+1}^{sl})$ from (25). Finally, for $\zeta_t = 1 - \alpha(\kappa_t^{sl})$, we get $\zeta_t > \zeta_{t+1}$ in (26). This finishes the proof of global superlinear convergence. The aggregated convergence rate is equal to (28). □

Similar results hold for Basic Tensor methods from (11) in general for $p \geq 2$. Next, we present the theorem for global superlinear convergence of Basic Tensor methods.

**Theorem 4.6** *For $\mu_q$-uniformly star-convex* (17) *function $f$ of degree $q \geq 2$ with $L_p$ - Lipschitz-continuous $p$-th derivative ($p \geq q \geq 2$)* (2), *Basic Tensor Method from* (11) *with $M_p \geq pL_p$ converges converges globally superlinearly as defined in* (15) *with $\zeta_{t,p} = 1 - \alpha_{t,p}^{sl}$*

$$f(x_{t+1}) - f^* \leq (1 - \alpha_{t,p}^{sl})(f(x_t) - f^*), \tag{29}$$

*where $\alpha_{t,p}^{sl}$ is such that*

$$h_{\kappa_{t,p}^{sl}}(\alpha_{t,p}^{sl}) = 0, \quad where \quad h_\kappa(\alpha) = \alpha^p \kappa + \alpha - 1, \quad \alpha_p^{low} = \min\left\{\frac{1}{2}; \frac{1}{2}\left(\frac{(p+1)!\mu}{q(M_p+L_p)D^{p-q+1}}\right)^{1/p}\right\}$$

$$and \quad \kappa_{t,p}^{sl} = \frac{(M_p+L_p)q^{(q+1)/q}}{(p+1)!\mu^{(q+1)/q}}(1-\alpha_p^{low})^{t/q}(f(x_0) - f(x^*))^{1/q}. \tag{30}$$

*The aggregated convergence rate for $T \geq 1$ equals to*

$$f(x_T) - f(x^*) \leq (f(x_0) - f(x^*))\prod_{t=1}^{T}(1 - \alpha_{t,p}^{sl}). \tag{31}$$

To sum up, we present a unified table for $\mu$-strongly (star-)convex functions.

| Method | Per-Iteration Rate $\alpha_t$ | Glob. Superlinear |
|---|---|---|
| Gradient Descent (Nesterov, 2004) | $\frac{\mu}{L_1}$ | ✗ |
| Cubic Newton Method (Nesterov, 2008) | $\left(\frac{\mu}{L_2D}\right)^{1/2}$ | ✗ |
| Basic Tensor Method (Doikov & Nesterov, 2022) | $\left(\frac{\mu}{L_pD^{p-1}}\right)^{1/(p+1)}$ | ✗ |
| Cubic Newton Method (NEW) | $\frac{\mu^{3/4}}{L_2^{1/2}}\left(1 - \left(\frac{\mu}{L_2D}\right)^{1/2}\right)^{-t/4}\Delta_0^{-1/4}$ | ✓ |
| Basic Tensor Method (NEW) | $\frac{\mu^{3/2p}}{L_2^{1/p}}\left(1 - \left(\frac{\mu}{L_pD^{p-1}}\right)^{1/p}\right)^{-t/2p}\Delta_0^{-1/2p}$ | ✓ |

Table 1: Comparison of per-iteration convergence for different basic methods, where $\Delta_0 = f(x_0) - f(x^*)$. To enhance clarity and simplicity, we removed universal constants and simplified (27) and (30) for the case where $\kappa_t \geq 1$.

We established the global superlinear convergence of Cubic Regularized Newton Method for $\mu$-strongly star-convex functions, as well as Basic Tensor Method for $\mu_q$-uniformly star-convex functions. Comprehensive details and proofs are provided in Appendix B.

## 5 CONCLUSION

**Limitations.** This paper primarily focuses on high-order methods which come with certain limitations. First of all, they have computational and memory limitations in high-dimensional spaces, due to the need for Hessian calculations. There are, however, approaches to overcome this, such as using first-order subsolvers or inexact Hessian approximations like Quasi-Newton approximations (BFGS, L-SR1). In this paper, we focus on the exact Hessian to analyze methods' peak performance.
Another limitation arises from the specific function classes and the theoretical results considered. Nonetheless, many of the proposed methods can be practically applied to a broader set of problems. For instance, the CRN performs competitively from general non-convex to strongly convex functions.

**Conclusion and Future work.** In the paper, we introduced *OPTAMI*, an open-source library designed to make high-order optimization methods more accessible and easier to experiment with. We plan to expand this library to cover a wider range of settings and optimization methods in the future.
In the first part of the paper, we proposed NATA, a practical acceleration technique. NATA employs a more aggressive schedule adaptation for $A_t$, enabling faster convergence. Our experimental results show that NATA significantly outperforms both basic and accelerated methods, including near-optimal and optimal methods. This opens up another interesting question: *Could other high-order methods be optimized by addressing practical issues that arise due to overly conservative theoretical guarantees?* Finally, we demonstrated that the basic high-order methods exhibit *global superlinear convergence* for $\mu$-strongly star-convex functions. This result is significant because it shows that high-order methods accelerate with each iteration, in stark contrast to first-order methods, which typically have a steady linear convergence rate. This raises intriguing questions: *Can global superlinear convergence be established for accelerated high-order methods as well? What is the best possible global per-iteration decrease that we can theoretically guarantee?*

## 6 ACKNOWLEDGEMENTS

The work of Dmitry Kamzolov in this paper has been partially funded by the Agence Nationale de la recherche under grant ANR-17-EURE-0010 (Investissement d'Avenir program), Toulouse School of Economics, Toulouse, France.
We sincerely thank the reviewers for their thoughtful feedback and valuable insights, which have helped improve this paper.

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

# A    RELATED WORKS

The origins of Newton method trace back to the foundational works on root-finding algorithms Newton (1687), Raphson (1697), Simpson (1740), and Bennett (1916). The next breakthrough in applying Newton method to optimization and proving its local quadratic convergence rates was done by Kantorovich (1948b;a; 1949; 1951b;a; 1956; 1957). Over the following decades, Newton's method have been studied in depth, modified and improved in works of Moré (1977) Griewank (1981); Nesterov & Polyak (2006). Today, Newton's method is widely used in industrial and scientific computing. For a more detailed history of Newton method, see Boris T. Polyak's paper (Polyak, 2007).

Recently, second-order methods have taken a new direction in development with the introduction of globally convergent methods achieving convergence rates of $O(T^{-2})$ (Nesterov & Polyak, 2006) and $O(T^3)$ (Nesterov, 2008) convergence rate, surpassing the performance of first-order methods (Nesterov, 2018). These advancements were later extended to higher-order (tensor) methods by Baes (2009). However, the tensor subproblem in these methods is nonconvex, leading to implementation challenges. This issue was addressed by the introduction of the (Accelerated) Tensor Method in Nesterov (2021b), which resolved the nonconvexity by increasing the scaling coefficient of the regularization term, making the subproblem convex. The basic $p$-th order Tensor Method achieves a rate of $O\left(T^{-p}\right)$, while the accelerated version improves this to $O\left(T^{-(p+1)}\right)$. Earlier work by Monteiro & Svaiter (2013) demonstrated that even faster convergence for second-order methods is possible with the Accelerated Proximal Extragradient method (A-HPE), achieving a rate of $\tilde{O}\left(T^{-7/2}\right)$. Lower bounds for second-order and higher-order methods of $\Omega\left(T^{-(3p+1)/2}\right)$ were established in (Arjevani et al., 2019; Nesterov, 2021b), demonstrating that the A-HPE method is nearly optimal for second-order convex optimization. Subsequently, three independent research groups (Gasnikov et al., 2019a; Bubeck et al., 2019; Jiang et al., 2019) extended the A-HPE framework to develop tensor methods with a convergence rate of $\tilde{O}\left(T^{-(3p+1)/2}\right)$, achieving near-optimal complexity for these higher-order methods. Truly optimal methods with a rate of $O\left(T^{-(3p+1)/2}\right)$ were later proposed in (Kovalev & Gasnikov, 2022; Carmon et al., 2022). Moreover, when assuming higher levels of smoothness, *second-order* methods (Nesterov, 2021c;a; Kamzolov, 2020; Doikov et al., 2024) have been shown to exceed the established lower complexity bounds for problems with Lipschitz-continuous Hessians. For an in-depth exploration of higher-order methods, see the review in (Kamzolov et al., 2023a).

Since second-order and higher-order methods generally incur greater computational costs due to the need for calculating higher-order derivatives, it is natural to consider inexact or stochastic algorithms to reduce these overheads. In convex optimization, several studies have explored globally convergent second-order methods with inexact Hessians (Ghadimi et al., 2017), higher-order methods with inexact and stochastic derivatives (Agafonov et al., 2024a; Kamzolov et al., 2020), and adaptive stochastic methods (Antonakopoulos et al., 2022). In (Agafonov et al., 2024b), a lower bound of $\Omega\left(\frac{\sigma_1}{\sqrt{T}} + \frac{\sigma_2}{T^2} + \frac{1}{T^{7/2}}\right)$ was established for stochastic globally convergent second-order methods, where $\sigma_1$ and $\sigma_2$ represent the variances of the stochastic gradients and Hessians, respectively. Additionally, the Accelerated Stochastic Cubic Newton method was introduced, achieving a convergence rate of $O\left(\frac{\sigma_1}{\sqrt{T}} + \frac{\sigma_2}{T^2} + \frac{1}{T^3}\right)$, which, to the best of our knowledge, represents the state-of-the-art result. Inexact second-order derivatives also studied for min-max problems and variational inequalities Lin et al. (2022); Agafonov et al. (2024c). Inexact second-order methods enable the use of Quasi-Newton Hessian approximations, which are well-regarded for their strong practical performance. Although classical Quasi-Newton (QN) methods are known for local superlinear convergence but lack global convergence, their integration with cubic regularization has led to globally convergent methods that also feature relatively inexpensive subproblem solutions (Kamzolov et al., 2023b; Scieur, 2023; Jiang et al., 2023). Second-order methods with inexact or stochastic derivatives also hold promise for distributed optimization Shamir et al. (2014); Reddi et al. (2016); Zhang & Lin (2015); Daneshmand et al. (2021); Agafonov et al. (2021); Dvurechensky et al. (2022); Agafonov et al. (2022), offering an effective way to manage the computational demands typically encountered in distributed settings. One actively developing direction relies on the constructions of Cubic Newton with explicit step in order to reduce the complexity of solving methods' subproblems Polyak (2009; 2017); Mishchenko (2023); Doikov & Nesterov (2023); Doikov et al. (2024); Hanzely et al. (2022).

# B  GLOBAL SUPERLINEAR CONVERGENCE

In this section, we show the theoretical global superlinear convergence of high-order methods ($p \geq 2$) for $\mu$-strongly star-convex functions.

**Theorem B.1** *For $\mu_q$-uniformly star-convex* (17) *function $f$ of degree $q \geq 2$ with $L_p$ - Lipschitz-continuous $p$-th derivative ($p \geq q \geq 2$) (2), Basic Tensor Method from* (11) *with $M_p \geq (p-1)L_p$ converges with the rate*

$$f(x_{t+1}) - f^* \leq (1 - \alpha_{t,p})(f(x_t) - f^*), \tag{32}$$

*for all $\alpha_{t,p}^* \geq \alpha_{t,p} \geq 0$ such that*

$$h_{\kappa_{t,p}}(\alpha_{t,p}) \leq 0 \text{ and } h_{\kappa_{t,p}}(\alpha_{t,p}^*) = 0, \text{ where } h_\kappa(\alpha) = \alpha^p \kappa + \alpha - 1 \text{ and } \kappa_{t,p} = \frac{q(M_p + L_p)\|x_t - x^*\|^{p-q+1}}{(p+1)!\mu}. \tag{33}$$

*This range includes the classical linear rate*

$$f(x_t) - f(x^*) \leq (1 - \alpha_p^{low})^t (f(x_0) - f(x^*)) \text{ for } \alpha_p^{low} = \min\left\{\frac{1}{2}; \frac{1}{2}\left(\frac{(p+1)!\mu}{q(M_p+L_p)D^{p-q+1}}\right)^{1/p}\right\} \tag{34}$$

*Proof.* We start the proof from an upper-bound (9)

$$f(x_{t+1}) \overset{(9)}{\leq} \Phi_{x_t,p}(x_{t+1}) + \frac{L_p}{(p+1)!}\|x_{t+1} - x_t\|^{p+1} \overset{(11)}{\leq} \min_{y\in\mathbb{R}^n}\left\{\Phi_{x_t,p}(y) + \frac{M_p}{(p+1)!}\|y - x_t\|^{p+1}\right\}$$

$$\overset{(9)}{\leq} \min_{y\in\mathbb{R}^n}\left\{f(y) + \frac{M_p+L_p}{(p+1)!}\|y - x_t\|^{p+1}\right\} \overset{y=x_t+\alpha_{t,p}(x^*-x_t)}{\leq} f((1-\alpha_{t,p})x_t + \alpha_{t,p}x^*) + \alpha_{t,p}^{p+1}\frac{M_p+L_p}{(p+1)!}\|x^* - x_t\|^{p+1}$$

$$\overset{(17)}{\leq} (1-\alpha_{t,p})f(x_t) + \alpha_{t,p}f(x^*) - \frac{\alpha_{t,p}(1-\alpha_{t,p})\mu}{q}\|x_t - x^*\|^q + \alpha_{t,p}^{p+1}\frac{M_p+L_p}{(p+1)!}\|x^* - x_t\|^{p+1}.$$

From the third inequality, we get that the method is monotone and $f(x_{t+1}) \leq f(x_t)$. Next, we subtract $f(x^*)$ from the both sides and get

$$f(x_{t+1}) - f(x^*) \leq (1-\alpha_{t,p})(f(x_t) - f(x^*)) - \frac{\alpha_{t,p}}{q}\|x_t - x^*\|^q \left((1-\alpha_t)\mu - \alpha_{t,p}^p \frac{q(M_p+L_p)}{(p+1)!}\|x_t - x^*\|^{p+1-q}\right),$$

If we choose $\alpha_t$ such that

$$\alpha_{t,p}^p \frac{q(M_p+L_p)}{(p+1)!}\|x_t - x^*\|^{p-q+1} + \mu\alpha_{t,p} - \mu \leq 0,$$

or equivalent version

$$\alpha_{t,p}^p \frac{q(M_p+L_p)}{(p+1)!\mu_q}\|x_t - x^*\|^{p-q+1} + \alpha_{t,p} - 1 \leq 0,$$

we get (32). To understand the solutions of such inequality, we present Lemma B.2 with the useful properties. From this Lemma, the convergence rate is well-defined as $0 < \alpha_{t,p}^* \leq 1$ from (36). As $\|x_t - x^*\| \leq D$ from (3) and $\alpha_p^{low} \leq \alpha_{t,p}^*$ by (36), we get the linear convergence rate (34). □

**Lemma B.2** *For $z > 0$, the solution $\alpha^*(z)$ of*

$$h_z(\alpha^*(z)) = 0, \qquad where \qquad h_z(\alpha) = \alpha^p z + \alpha - 1, \tag{35}$$

*has the next constant lower and upper-bound*

$$\min\left\{1, \frac{1}{z^{1/p}}\right\} > \alpha^*(z) > \min\left\{\frac{1}{2}; \frac{1}{2z^{1/p}}\right\}. \tag{36}$$

*This bounds show that, for $z \geq 1$, the solution $\alpha^*(z)$ is similar to $z^{-1/p}$ up to a constant factor as $z^{-1/p} > \alpha^*(z) > 0.5z^{-1/p}$.*

*For $z \leq 1$, we get the next improved upper and lower-bound*

$$1 - \frac{z}{p+1} \geq \alpha^*(z) \geq 1 - z, \tag{37}$$

*which means that for $z \to +0$ we have $\alpha^*(z) \to 1$.*

*The solution $\alpha^*(z)$ is monotonically decreasing*

$$\forall z, y > 0: \quad z < y \quad \Rightarrow \quad \alpha^*(z) > \alpha^*(y). \tag{38}$$

*Proof.* We start the proof from the upper-bound inequality. Note, the function $h_z(\alpha) = \alpha^p z + \alpha - 1$ is monotonically increasing for $\alpha \geq 0$, as $h_z(\alpha)' = p\alpha^{p-1}z + 1 > 0$. As $h_z(0) = -1$ and $h_z(1) = z > 0$, we get that the solution is unique and $\alpha^*(z) \in [0; 1]$. Next, for $\alpha = \frac{1}{z^{1/p}}$, we have

$$h_z(\tfrac{1}{z^{1/p}}) = \tfrac{1}{z}z + \tfrac{1}{z^{1/p}} - 1 = \tfrac{1}{z^{1/p}} > 0,$$

which means that $\min\left\{1, \frac{1}{z^{1/p}}\right\} > \alpha^*(z)$ and we proved the upper-bound.

Next, we move to the lower-bound inequality. For $p \geq 2$, $z \geq 1$ and $\alpha = \frac{1}{2z^{1/p}}$, we have

$$h_z(\tfrac{1}{2z^{1/p}}) = \tfrac{1}{2^p z}z + \tfrac{1}{2z^{1/p}} - 1 = \tfrac{1}{2^p} + \tfrac{1}{2z^{1/p}} - 1 \leq \tfrac{1}{2^p} + \tfrac{1}{2} - 1 < 0,$$

where the first inequality is coming from $z \geq 1$. The second part of lower-bound holds for $0 < z < 1$ because

$$h_z(\tfrac{1}{2}) = \tfrac{1}{2^p}z + \tfrac{1}{2} - 1 < \tfrac{1}{2^p} - \tfrac{1}{2} < 0.$$

We proved (35). Now, to understand the behavior of $\alpha^*(z)$ for $0 < z \leq 1$, we improve the upper and lower-bound for $0 < z \leq 1$. For $0 < z \leq 1$ and $\alpha = 1 - z$, we get the improved lower-bound

$$h_z(1 - z) = (1 - z)^p z + 1 - z - 1 = (1 - z)^p z - z = ((1 - z)^p - 1)z < 0.$$

For $p \geq 2$, $0 < z \leq 1$ and $\alpha = 1 - \frac{z}{p+1}$, we get the improved upper-bound

$$h_z\left(1 - \tfrac{z}{p+1}\right) = \left(1 - \tfrac{z}{p+1}\right)^p z + 1 - \tfrac{z}{p+1} - 1 = \left(1 - \tfrac{z}{p+1}\right)^p z - \tfrac{z}{p+1}$$

$$\geq \left(\left(1 - \tfrac{1}{p+1}\right)^p - \tfrac{1}{p+1}\right)z = \left(\tfrac{p^p - (p+1)^{p-1}}{(p+1)^p}\right)z > 0, \tag{39}$$

where to use the last inequity or $p \geq 2$ we need to use some additional analysis. We introduce an additional function and its derivatives

$$s(x) = x\log(x) - (x - 1)\log(x + 1),$$

$$s(x)' = \tfrac{2}{1+x} + \log\left(1 - \tfrac{1}{1+x}\right),$$

$$s(x)'' = \tfrac{1-x}{x(1+x)^2}.$$

It is clear that $s(x)'' < 0$ for $x > 1$. It means that $s(x)'$ is monotonically decreasing. $s(1)' = 1 - \log(2) > 0$ and the limit $\lim_{x \to +\infty} s(x)' = 0$, hence $s(x)' \geq 0$ and $s(x)$ is a monotonically increasing function. $s(1) = 0$, hence $s(x) > 0$ for $x > 1$ and finally $x^x > (x + 1)^{x-1}$ for $x > 1$, which proves (39) and finishes the proof of the improved upper-bound (37).

Finally, we show that the solution $\alpha^*(z)$ is monotonically decreasing with $z$. Let $0 < z < y$ and $\alpha^*(z)$ and $\alpha^*(y)$ are such that $h_z(\alpha^*(z)) = 0$ and $h_y(\alpha^*(y)) = 0$, then

$$\alpha^*(z)^p y + \alpha^*(z) - 1 \overset{(35)}{=} \alpha^*(z)^p y + 1 - \alpha^*(z)^p z - 1 = \alpha^*(z)^p(y - z) > 0,$$

which proves that $\alpha^*(z) > \alpha^*(y)$ and hence the solution $\alpha^*(z)$ is monotonically decreasing. $\square$

Now, we proceed to the second theorem to establish the global superlinear convergence of high-order methods. The key idea behind the proof is to observe that $|x_t - x^*|$ in (33) decreases for $\mu_q$-uniformly star-convex functions. This allows us to notice the fact that $\kappa_{t,p}$ is also decreasing, hence $\alpha_{t,p}$ increases according to (38), ultimately leading to superlinear convergence.

**Theorem B.3** (*Copy of Theorem 4.6*) *For $\mu_q$-uniformly star-convex* (17) *function $f$ of degree $q \geq 2$ with $L_p$ - Lipschitz-continuous $p$-th derivative ($p \geq q \geq 2$)* (2)*, Basic Tensor Method from* (11) *with $M_p \geq (p - 1)L_p$ converges converges globally superlinearly as defined in* (15) *with $\zeta_{t,p} = 1 - \alpha_{t,p}^{sl}$*

$$f(x_{t+1}) - f^* \leq (1 - \alpha_{t,p}^{sl})(f(x_t) - f^*), \tag{40}$$

*where $\alpha_{t,p}^{sl}$ is such that*

$$h_{\kappa_{t,p}^{sl}}(\alpha_{t,p}^{sl}) = 0, \quad where \quad h_\kappa(\alpha) = \alpha^p \kappa + \alpha - 1, \quad \alpha_p^{low} = \min\left\{\tfrac{1}{2}; \tfrac{1}{2}\left(\tfrac{(p+1)!\mu}{q(M_p+L_p)D^{p-q+1}}\right)^{1/p}\right\}$$

$$and \quad \kappa_{t,p}^{sl} = \tfrac{(M_p+L_p)q^{(q+1)/q}}{(p+1)!\mu^{(q+1)/q}}(1 - \alpha_p^{low})^{t/q}(f(x_0) - f(x^*))^{1/q}. \tag{41}$$

*The aggregated convergence rate for $T \geq 1$ equals to*

$$f(x_T) - f(x^*) \leq (f(x_0) - f(x^*))\prod_{t=1}^T (1 - \alpha_{t,p}^{sl}). \tag{42}$$

*Proof.* From $\mu$-uniform star-convexity (17), we can upper-bound $\|x_t - x^*\|$ in (33) by

$$\|x_t - x^*\| \leq \left(\frac{q}{\mu}\left(f(x_t) - f(x^*)\right)\right)^{1/q} \overset{(34)}{\leq} \left(\frac{q}{\mu}\left((1-\alpha^{low})^t(f(x_0) - f(x^*))\right)\right)^{1/q}.$$

So, we got that $\|x_t - x^*\|$ is linearly decreasing to zero. From that, we get a new superlinear $\alpha_{t,p}^{sl} \leq \alpha_{t,p}^{*}$ from (41). As $\kappa_t^{sl}$ is getting smaller within each iteration $\kappa_{t,p}^{sl} > \kappa_{t+1,p}^{sl}$, we get that $\alpha(\kappa_{t,p}^{sl}) < \alpha(\kappa_{t+1,p}^{sl})$ from (38). Finally, for $\zeta_{t,p} = 1 - \alpha(\kappa_{t,p}^{sl})$, we get $\zeta_{t,p} > \zeta_{t+1,p}$ in (40). This finishes the proof of global superlinear convergence. The aggregated convergence rate equals to (42). $\square$

# C SUBSOLVERS

## C.1 SUBSOLVER FOR BASIC TENSOR METHOD

In this section, we introduce the subsolver, called Bregman Distance Gradient Method (BDGM), for the Basic Tensor Method of order $p = 3$ (12):

$$x_{t+1} = x_t + \underset{h \in \mathbb{E}}{\operatorname{argmin}} \left\{ f(x_t) + \nabla f(x_t)\,[h] + \tfrac{1}{2}\nabla^2 f(x_t)\,[h]^2 + \tfrac{1}{6}D^3 f(x_t)\,[h]^3 + \tfrac{M_3}{24}\|h\|^4 \right\}. \quad (43)$$

The first effective subsolver was introduced by Nesterov in (Nesterov, 2021b, Section 5) and later improved in (Nesterov, 2021c). Next, we describe the BDGM subsolver by following the (Nesterov, 2021c).

**Relatively inexact $p$-th order solution.** First, we introduce the relatively inexact $p$-th order solution of (11)

$$\mathcal{N}_{M_p}^{\gamma}(x) = \left\{ y \in \mathbb{E} : \|\nabla\Omega_{x,M_p}(y)\|_* \leq \gamma\|\nabla f(y)\|_* \right\}, \quad (44)$$

where $\gamma \in [0,1)$ is an accuracy parameter. Then from (Nesterov, 2021c, Theorem 2.1), for $\gamma$ and $M_p$ such that $\gamma + \frac{L_p}{M_p} \leq \frac{1}{p}$ any point $y \in \mathcal{N}_{M_p}^{\gamma}(x)$ satisfies

$$f(x) - f(y) \geq c_{\gamma,M_p} \|\nabla f(y)\|_*^{\frac{p+1}{p}}, \quad \text{where} \quad c_{\gamma,M_p} = \left[ \frac{(1-\gamma)p!}{L_p + M_p} \right]^{\frac{1}{p}}.$$

Note, that for the exact solution, we get the same improvement guarantee with $\gamma = 0$. For $p = 3$ and (43), we choose $\gamma = 1/6$ and $M_3 = 6L_3$, then $\mathcal{N}_{L_3}(x) = \mathcal{N}_{L_3}^{1/6}(x)$ and the method

$$x_{t+1} \in \mathcal{N}_{L_3}(x_t)$$

converge with the same rate up to a constant as an exact version (Nesterov, 2021c, Theorem 2.2). Note, $M_3 \geq 3L_3$ is also required for the convexity of the subproblem (43). In our implementation, all third-order basic steps are solved with this relative inexactness and $M_3 = 6L_3$. This approach creates practical and parameter-free stopping criteria for the subproblem solvers.

**Relative smoothness and relative strong convexity.** Now, we move on to the concept of relative smoothness and relative strong convexity proposed in (Lu et al., 2018). Similarly to classical smoothness and strong convexity, we say that function $\phi(h)$ is relatively $L_\rho$-smooth and relatively $\mu_\rho$ strongly convex with respect to scaling function $\rho(h)$ if

$$\mu_\rho \nabla^2 \rho(h) \preceq \nabla^2 \phi(h) \preceq L_\rho \nabla^2 \rho(h).$$

In classical regime, $\rho(h) = \frac{1}{2}\|h\|^2$ and $\nabla^2 \rho(h)$ is an identity matrix. For the scaling function $\rho(h)$, we introduce its Bregman distance

$$\beta_\rho(h,y) = \rho(y) - \rho(h) - \langle \nabla\rho(h), y - h \rangle.$$

Now the gradient method with respect to this Bregman distance is called Bregman Distance Gradient Method (BDGM) and has the next form

$$h_{k+1} = \underset{y \in \mathbb{E}}{\operatorname{argmin}} \left\{ \langle \nabla\phi(h_k), y - h_k \rangle + 2L_\rho\beta_\rho(h_k, y) \right\}.$$

The convergence rate of such method is $O\left( \frac{L_\rho}{\mu_\rho} \log\left( \frac{\phi(h_0) - \phi(h^*)}{\varepsilon} \right) \right)$.

**Bregman Distance Gradient Method (BDGM) for** (43). Let's apply this approach to the solution of subproblem (43) with $M_3 = 6L_3$. In (Nesterov, 2021c, Section 4), it was shown that the subproblem function $\phi(h) = \nabla f(x_t)[h] + \frac{1}{2}\nabla^2 f(x_t)[h]^2 + \frac{1}{6}D^3 f(x_t)[h]^3 + \frac{L_3}{4}\|h\|^4$ is relatively smooth and relatively strongly convex with respect to

$$\rho(h) = \tfrac{1}{2}\nabla^2 f(x_t)[h]^2 + \tfrac{L_3}{4}\|h\|^4$$

with constants $L_\rho = 1 + \frac{1}{\sqrt{2}}$ and $\mu_\rho = 1 - \frac{1}{\sqrt{2}}$. It means that the method has an incredibly fast convergence rate $O\left(\frac{\sqrt{2}+1}{\sqrt{2}-1}\log\left(\frac{\phi(h_0)-\phi(h^*)}{\varepsilon}\right)\right)$. The details and more formal convergence results are presented in (Nesterov, 2021c).

Now, we present the explicit formulation of the BDGM for (43). First, we have the general form

$$h_{k+1} = \underset{y\in\mathbb{E}}{\arg\min}\left\{\langle\nabla\phi(h_k), y - h_k\rangle + 2L_\rho\beta_\rho(h_k, y)\right\}. \tag{45}$$

Let us calculate $\nabla\phi(h_k)$ first. It equals to

$$\nabla\phi(h_k) = \nabla f(x_t) + \nabla^2 f(x_t)h_k + \tfrac{1}{2}D^3 f(x_t)[h_k]^2 + L_3\|h_k\|^2 h_k.$$

In (Nesterov, 2021c), the universal approximation for $D^3 f(x_t)[h_k]^2$ is presented by using the finite differences approach. However, in practice, we recommend using autogradient computation of $D^3 f(x_t)[h_k]^2$ if it is possible. The computation by autogradient is much more precise while having the same computational complexity. The computation complexity of $D^3 f(x_t)[h_k]^2$ by autogradient is similar to calculating three gradients as $D^3 f(x)[h]^2 = \nabla_x(\nabla^2 f(x)[h]^2) = \nabla(\nabla\{\nabla f(x)[h]\}[h])$. Also, autogradient computations are commonly used in modern frameworks such as PyTorch, Jax, and others. So, essentially we still have access to third-order information but with the complexity of a gradient computation.

Now, let us calculate explicit $\beta_\rho(h_k, y)$

$$\beta_\rho(h_k, y) = \rho(y) - \rho(h) - \langle\nabla\rho(h), y - h\rangle$$
$$= \tfrac{1}{2}\nabla^2 f(x_t)[y]^2 + \tfrac{L_3}{4}\|y\|^4 - \tfrac{1}{2}\nabla^2 f(x_t)[h_k]^2 - \tfrac{L_3}{4}\|h_k\|^4$$
$$- \left\langle\nabla^2 f(x_t)[h_k] + L_3\|h_k\|^2 h_k, y - h_k\right\rangle.$$

Note, that the constant terms are useless for finding the argminimum in (45), hence we can remove them. We also can divide all parts of (45) by $2L_\rho = 2 + \sqrt{2}$ for simplicity and unite the linear parts together

$$g_k = \tfrac{1}{2+\sqrt{2}}\nabla\phi(h_k) - \nabla^2 f(x_t)[h_k] - L_3\|h_k\|^2 h_k$$
$$= \tfrac{2-\sqrt{2}}{2}\left(\nabla f(x_t) + \nabla^2 f(x_t)h_k + \tfrac{1}{2}D^3 f(x_t)[h_k]^2 + L_3\|h_k\|^2 h_k\right) - \nabla^2 f(x_t)[h_k] - L_3\|h_k\|^2 h_k$$
$$= \tfrac{2-\sqrt{2}}{2}\left(\nabla f(x_t) + \tfrac{1}{2}D^3 f(x_t)[h_k]^2\right) - \tfrac{\sqrt{2}}{2}\left(\nabla^2 f(x_t)[h_k] + L_3\|h_k\|^2 h_k\right)$$

So, we finally get the next explicit BDGM step

$$h_{k+1} = \underset{y\in\mathbb{E}}{\arg\min}\left\{\langle g_k, y\rangle + \tfrac{1}{2}\nabla^2 f(x_t)[y]^2 + \tfrac{L_3}{4}\|y\|^4\right\}. \tag{46}$$

This step doesn't require the computation of a full third-order derivative and is similar to the Cubic Regularized Newton step. Hence, we count it as a second-order method. So, the total complexity of Basic Tensor Method for convex functions is $\tilde{O}\left(\frac{L_3 D^4}{T^3}\right)$ steps of (46), where $\tilde{O}(\cdot)$ means number of iterations up to a logarithmic factor.

**Inner subsolver for** (46). The last part is to solve (46). We solve it similarly to the Cubic Regularized Step by ray-search with eigenvalue decomposition (EVD). First, we apply eigenvalue decomposition to $\nabla^2 f(x_t)$

$$\nabla^2 f(x_t) = USU^\top, \tag{47}$$

where $S \in \mathbb{R}^{d \times d}$ is a diagonal matrix with eigenvalues and $U \in \mathbb{R}^{d \times d}$ is an orthoganal matrix such that $UU^\top = I$. Then, we denote $v = U^\top y$ and $\hat{g} = U^\top g_k$. Now we can formulate a dual one-dimensional problem.

$$
\begin{aligned}
\min_{y \in \mathbb{E}} & \left\{ \langle g_k, y \rangle + \tfrac{1}{2} \left\langle \nabla^2 f(x_k) y, y \right\rangle + \tfrac{L_3}{4} \|y\|^4 \right\} \\
= \min_{y \in \mathbb{E}} & \left\{ \left\langle U^\top g_k, U^\top y \right\rangle + \tfrac{1}{2} \left\langle USU^\top y, y \right\rangle + \tfrac{L_3}{4} \|U^\top y\|^4 \right\} \\
= \min_{v \in \mathbb{E}} & \left\{ \langle \tilde{g}, v \rangle + \tfrac{1}{2} \langle Sv, v \rangle + \tfrac{L_3}{4} \|v\|^4 \right\} \\
= \min_{v \in \mathbb{E}} \max_{\tau \geq 0} & \left\{ \langle \tilde{g}, v \rangle + \tfrac{1}{2} \langle Sv, v \rangle + \tfrac{\sqrt{2L_3}}{2} \|v\|^2 \tau - \tfrac{1}{2}\tau^2 \right\} \\
= \max_{\tau \geq 0} \min_{v \in \mathbb{E}} & \left\{ \langle \tilde{g}, v \rangle + \tfrac{1}{2} \langle Sv, v \rangle + \tfrac{\sqrt{2L_3}}{2} \|v\|^2 \tau - \tfrac{1}{2}\tau^2 \right\} \\
= \max_{\tau \geq 0} & \left\{ -\tfrac{1}{2} \left\langle \left( S + \tau \sqrt{2L_3} \right)^{-1} \tilde{g}, \tilde{g} \right\rangle - \tfrac{1}{2}\tau^2 \right\},
\end{aligned}
\tag{48}
$$

where $\tau^* = \frac{\sqrt{2L_3}}{2}\|v\|^2$ for the third equality and $v = -\left( S + \tau \sqrt{2L_3} \right)^{-1} \tilde{g}$ in the last equality. By solving (48) with one-dimensional ray-search, we find optimal $\tau^*$ then we can calculate $v$ and $y$, which we found the solution for subproblem (46). In our code, we use eigenvalue decomposition for efficiency of the ray-search, but it is also possible to just inverse the regularized matrix multiple times in (48) or apply some efficient first-order method for quadratic problems such as conjugate gradient.

To finalize, in this section we presented the subsolver which allows us to efficiently implement the Basic Tensor Method for $p = 3$ with the complexity same up to a logarithmic factor as Cubic Regularized Newton Method.

# D    METHODS

## D.1    NESTEROV ACCELERATED TENSOR METHODS

In this section, we present Nesterov Acceleration for tensor methods proposed in (Nesterov, 2021b;c). First, let us introduce the main parts of the method. The key part of such acceleration is the estimated sequences technique. It is based on linear approximations of function $f(x)$ in a sequence of points $x_t$, which allows to construct the estimating function $\psi_t(x)$ for a scaling sequence $a_t \in \mathbb{R}_+$:

$$
\psi_{t+1}(z) = \psi_t(z) + a_{t+1}\left( f(x) + \langle \nabla f(x), z - x \rangle \right), \quad \text{where} \quad \psi_0(z) = \tfrac{1}{p+1}\|z - x_0\|^{p+1}. \tag{49}
$$

Additionally, we introduce the sequence

$$
A_{t+1} = A_t + a_t. \tag{50}
$$

Now, we are ready to present the accelerated method.

---

**Algorithm 3** Nesterov Accelerated Tensor Method

---

1: **Input:** $x_0$ is starting point; constant $L_p$, total number of iterations $T$, and sequence $A_t$, where $A_0 = 0$.
2: Set objective function
$$
\psi_0(z) = \tfrac{1}{p+1}\|z - x_0\|^{p+1}
$$
3: **for** $t \geq 0$ **do**
4:     Choose $y_t = \frac{A_t}{A_{t+1}} x_t + \frac{a_{t+1}}{A_{t+1}} v_t$
5:     Compute $x_{t+1} \in \mathcal{N}_{L_p}(y_t)$
6:     Compute $a_{t+1} = A_{t+1} - A_t$
7:     Update $\psi_{t+1}(x) = \psi_t(z) + a_{t+1}[f(x_{t+1}) + \langle \nabla f(x_{t+1}), z - x_{t+1} \rangle]$.
8:     Compute $v_{t+1} = \operatorname{argmin}_{z \in \mathbb{E}} \psi_{t+1}(z)$
9: **end for**
10: **return** $x_{T+1}$

---

For the convergence results, the sequence $A_t$ should be defined in the following way

$$A_t = \frac{\nu_p}{L_p} t^{p+1}, \quad where \quad \nu_p = \frac{2p-1}{(p+1)(2p+1)} \cdot \frac{(p-1)!}{(2p)^p}. \tag{51}$$

Then, $a_{t+1} = \frac{\nu_p}{L_p} \left((t+1)^{p+1} - t^{p+1}\right)$. With such parameters, we can present the convergence theorem from (Nesterov, 2021c, Theorem 2.3)

**Theorem D.1** *Let sequence $\{x_t\}_{t\geq 0}$ be generated by method 3. Then, for any $T \geq 1$, we have*

$$f(x_T) - f(x^*) \leq O\left(\frac{L_p R^{p+1}}{T^{p+1}}\right).$$

## D.2 NESTEROV ACCELERATED TENSOR METHOD WITH $A_t$-ADAPTATION (NATA)

In this subsection, we present the proof of Theorem 3.1

---

**Algorithm 4** Nesterov Accelerated Tensor Method with $A_t$-Adaptation (NATA)

---

1: **Input:** $x_0 = v_0$ is starting point, constant $M_p$, total number of iterations $T$, $\tilde{A}_0 = 0$, $\nu^{\min} = \nu_p$, $\nu^{\max} \geq \nu_p$ is a maximal value of $\nu$, $\theta > 1$ is a scaling parameter for $\nu$, and $\nu_0 = \nu^{\max}\theta$ is a starting value of $\nu$.
2: Set objective function

$$\psi_0(z) = \frac{1}{p+1}\|z - x_0\|^{p+1}$$

3: **for** $t \geq 0$ **do**
4:     **repeat**
5:       $\nu^t = \max\left\{\frac{\nu^t}{\theta}, \nu_{\min}\right\}$
6:       $\tilde{a}_{t+1} = \frac{\nu^t}{L_p}((t+1)^{p+1} - t^{p+1})$ and $\tilde{A}_{t+1} = \tilde{A}_t + \tilde{a}_{t+1}$
7:       $y_t = \frac{\tilde{A}_t}{\tilde{A}_{t+1}} x_t + \frac{\tilde{a}_{t+1}}{\tilde{A}_{t+1}} v_t$
8:       $x_{t+1} = \mathcal{N}_{L_p}(y_t)$
9:       $\psi_{t+1}(z) = \psi_t(z) + \tilde{a}_{t+1}[f(x_{t+1}) + \langle\nabla f(x_{t+1}), z - x_{t+1}\rangle]$
10:       $v_{t+1} = \operatorname{argmin}_{z \in \mathbb{E}} \psi_{t+1}(z)$
11:     **until** $\psi_{t+1}(v_{t+1}) \geq \tilde{A}_{t+1}f(x_{t+1})$
12:     $\nu^{t+1} = \min\left\{\nu^t\theta^2, \nu_{\max}\right\}$
13: **end for**
14: **return** $x_{T+1}$

---

**Theorem D.2** *(Copy of Theorem 3.1) For convex function $f$ with $L_p$-Lipschitz-continuous $p$-th derivative, to find $x_T$ such that $f(x_T) - f(x^*) \leq \varepsilon$, it suffices to perform no more than $T \geq 1$ iterations of the Nesterov Accelerated Tensor Method with $A_t$-Adaptation (NATA) with $M_p \geq pL_p$ (Algorithm 2), where*

$$T = O\left(\left(\frac{L_p R^{p+1}}{\varepsilon}\right)^{\frac{1}{p+1}} + \log_\theta\left(\frac{\nu^{\max}}{\nu^{\min}}\right)\right). \tag{52}$$

*Proof.*

Let us present the convergence analysis of Algorithm 2. The proof is based on the proof from (Nesterov, 2021c).

First of all, by convexity and definition of $\psi_t(x)$, it is easy to show that

$$\psi_t(x^*) \leq \tilde{A}_t f(x^*) + \frac{1}{p+1}\|x^* - x_0\|^{p+1}. \tag{53}$$

Now, let us assume that the condition on Line 10 is satisfied for every step. Then, we get

$$\tilde{A}_t f(x_t) \leq \psi_t(v_t) \leq \psi_t(x^*) \leq \tilde{A}_t f(x^*) + \frac{1}{p+1}\|x^* - x_0\|^{p+1}, \tag{54}$$

where in the second inequality we use the definition of $v_t$. Next, by simple calculations, we get the convergence result

$$f(x_t) - f(x^*) \leq \frac{\|x^* - x_0\|^{p+1}}{(p+1)\tilde{A}_t}. \tag{55}$$

From that inequality, one can see that the larger $\tilde{A}_t$ means the faster convergence. That is the reason, we want to have a more aggressive $\tilde{a}_t$ and start the search of $\nu$ from the maximal value. Now, we need to show that the condition in Line 10 is always can be satisfied.

Let us prove it by induction of the following relation:

$$\psi_t^* = \psi_t(v_t) \geq \tilde{A}_t f(x_t), \quad t \geq 0. \tag{56}$$

For $t = 0$, we have $\psi_0^* = 0$ and $A_0 = 0$. Hence, (56) is valid.

Assume it is valid for some $t \geq 0$. Then,

$$\psi_{t+1}^* = \psi_t(v_{t+1}) + \tilde{a}_{t+1}\left(f(x_{t+1}) + \langle \nabla f(x_{t+1}), v_{t+1} - x_{t+1}\rangle\right)$$
$$\geq \psi_t^* + \frac{1}{(p+1)2^{p-1}}\|v_{t+1} - v_t\|^{p+1} + \tilde{a}_{t+1}\left(f(x_{t+1}) + \langle \nabla f(x_{t+1}), v_{t+1} - x_{t+1}\rangle\right),$$

where the last inequality is coming from uniform convexity of $\|\cdot\|^{p+1}$. Now, we can use the structure of the method in previous inequality and get

$$\psi_{t+1}^* - \frac{1}{(p+1)2^{p-1}}\|v_{t+1} - v_t\|^{p+1} \overset{(56)}{\geq} \tilde{A}_t f(x_t) + \tilde{a}_{t+1}\left(f(x_{t+1}) + \langle \nabla f(x_{t+1}), v_{t+1} - x_{t+1}\rangle\right)$$
$$\geq \tilde{A}_{t+1}f(x_{t+1}) + \langle \nabla f(x_{t+1}), \tilde{a}_{t+1}(v_{t+1} - x_{t+1}) + \tilde{A}_t(x_t - x_{t+1})\rangle$$
$$= \tilde{A}_{t+1}f(x_{t+1}) + \langle \nabla f(x_{t+1}), \tilde{a}_{t+1}(v_{t+1} - v_t) + \tilde{A}_{t+1}(y_t - x_{t+1})\rangle,$$

where, for the second inequality, we use convexity and, for the last equality, we use the definition of $y_t$ from Line 6 of the Algorithm 2.

Further, we use inequality $\frac{\alpha}{p+1}\tau^{p+1} - \beta\tau \geq -\frac{p}{p+1}\alpha^{-1/p}\beta^{(p+1)/p}, \quad \tau \geq 0$, for all $x \in \mathbb{E}$ and we have

$$\frac{1}{(p+1)2^{p-1}}\|v_{t+1} - v_t\|^{p+1} + \tilde{a}_{t+1}\langle \nabla f(x_{t+1}), v_{t+1} - v_t\rangle \geq -\frac{p}{p+1}2^{\frac{p-1}{p}}\left(\tilde{a}_{t+1}\|\nabla f(x_{t+1})\|_*\right)^{\frac{p+1}{p}}. \tag{57}$$

Next, for $x_{t+1} \in \mathcal{N}_{L_p}(y_k)$, from (Nesterov, 2021c, Theorem 2.1), we get

$$\langle \nabla f(x_{t+1}), y_t - x_{t+1}\rangle \geq c_p\|\nabla f(x_{t+1})\|_*^{\frac{p+1}{p}},$$

where $c_p = \left[\frac{2p-1}{2p(2p+1)}\frac{p!}{L_p}\right]^{1/p}$ for relative inexact $p$-th order solution.

Putting all these inequalities together, we obtain

$$\psi_{t+1}^* \geq \tilde{A}_{t+1}f(x_{t+1}) - \frac{p}{p+1}2^{\frac{p-1}{p}}\left(\tilde{a}_{t+1}\|\nabla f(x_{t+1})\|_*\right)^{\frac{p+1}{p}} + \tilde{A}_{t+1}c_p\|\nabla f(x_{t+1})\|_*^{\frac{p+1}{p}}$$
$$= \tilde{A}_{t+1}f(x_{t+1}) + \|\nabla f(x_{t+1})\|_*^{\frac{p+1}{p}}\left(\tilde{A}_{t+1}c_p - \frac{p}{p+1}2^{\frac{p-1}{p}}\tilde{a}_{t+1}^{\frac{p+1}{p}}\right).$$

Finally, by the choice of $\nu^t$ in Algorithm 2, $\nu^t \geq \nu_p$ and $\tilde{a}_{t+1} \geq a_{t+1}$, where $a_{t+1} = \frac{\nu_p}{L_p}((t+1)^{p+1} - t^{p+1})$ is the theoretical value of $a_{t+1}$. Hence, $\tilde{A}_{t+1} \geq A_{t+1}$, where $A_{t+1} = \frac{\nu_p}{L_p}(t+1)^{p+1}$ is the theoretical value of $A_{t+1}$. So, in the final inequality, we prove that there exists $\nu^t = \nu_p$ such that

$$\tilde{A}_{t+1}c_p \geq A_{t+1}c_p \geq \frac{p}{p+1}2^{\frac{p-1}{p}}a_{t+1}^{\frac{p+1}{p}},$$

where the last inequality holds from (Nesterov, 2021c, Equation 25). Thus, we have proved the induction step.

The search of $\nu^t$ takes maximal total of $\log_\theta\left(\frac{\nu_t^{\max}}{\nu_t^{\min}}\right) + T$ additional steps, where $\nu_t^{\max} = \max_{t \in [0;T]} \nu^t \leq \nu^{\max}$ and $\nu_t^{\min} = \min_{t \in [0;T]} \nu^t \geq \nu^{\min} = \nu_p$. The $T$ term in the sum is coming from Line 11 in Algorithm 2. If we want to make the Algorithm less aggressive, we can remove this Line then $\nu^t$ will only decrease.

The total number of iterations hence is equal to $T = O\left(\left(\frac{L_p R^{p+1}}{\varepsilon}\right)^{\frac{1}{p+1}} + \log_\theta\left(\frac{\nu_t^{\max}}{\nu_t^{\min}}\right)\right)$, which

finishes the proof. □

### D.3 NEAR-OPTIMAL TENSOR METHODS AND HYPERFAST SECOND-ORDER METHOD

**Near-optimal Tensor methods.** Monteiro & Svaiter (2013) demonstrated that the global convergence rate of second-order methods can be further improved from $O\left(\varepsilon^{-1/3}\right)$ to $O\left(\varepsilon^{-2/7}\log\left(1/\varepsilon\right)\right)$. This improvement was achieved through the development of the Accelerated Hybrid Proximal Extragradient (A-HPE) framework, which, when combined with a trust-region Newton-type method, resulted in the Accelerated Newton Proximal Extragradient (A-NPE) method that achieves the improved rate. A lower bound of $O\left(\varepsilon^{-2/7}\right)$ was established by Arjevani et al. (2019), rendering that the A-NPE method is nearly optimal.

Near-optimal tensor methods Gasnikov et al. (2019a); Bubeck et al. (2019); Jiang et al. (2019), with a convergence rate of $O\left(\varepsilon^{-2/(3p+1)}\log\left(1/\varepsilon\right)\right)$, are based on the A-HPE framework. Similar to A-HPE, these tensor methods require an additional binary search procedure at each iteration. The cost of these procedures introduces an extra $O(\log(1/\varepsilon))$ factor in the overall convergence rate.

---

**Algorithm 5** Inexact $p$-th order Near-optimal Accelerated Tensor Method (Kamzolov, 2020, Algorithm 1)

---

1: **Input:** $x_0 = v_0$ is starting point, constants $M_p$, $\gamma \in [0,1)$, total number of iterations $T$, $A_0 = 0$.

2: Set $A_0 = 0$, $x_0 = v_0$
3: **for** $t \geq 0$ **do**
4:    Compute a pair $\lambda_{t+1} > 0$ and $x_{t+1} \in \mathbb{R}^n$ such that

$$\frac{1}{2} \leq \lambda_{t+1} \frac{M_p \cdot \|x_{t+1} - y_t\|^{p-1}}{(p-1)!} \leq \frac{p}{p+1} \tag{58}$$

   where

$$x_{t+1} \in \mathcal{N}_{p,M_p}^\gamma(y_t) \tag{59}$$

   and

$$a_{t+1} = \frac{\lambda_{t+1} + \sqrt{\lambda_{t+1}^2 + 4\lambda_{t+1}A_t}}{2} \ , \ A_{t+1} = A_t + a_{t+1} \ , \text{ and } y_t = \frac{A_t}{A_{t+1}}x_t + \frac{a_{t+1}}{A_{t+1}}v_t \ . \tag{60}$$

5:    Update $v_{t+1} = v_t - a_{t+1}\nabla f(x_{t+1})$
6: **end for**
7: **return** $y_K$

---

One version of the near-optimal tensor methods is presented in Algorithm 5. This version was initially proposed by Bubeck et al. (2019) and later improved by Kamzolov (2020), who introduced the handling of inexact solution to subproblem (59). Note that line (4) of Algorithm 5 requires finding the pair $(x_{t+1}, \lambda_{t+1})$, which cannot be done explicitly. Specifically, $\lambda_{t+1}$ depends on $x_{t+1}$ via (58), which in turn depends on $y_t$ through (59). Furthermore, $y_t$ depends on $a_{t+1}$, which itself depends on $\lambda_{t+1}$ as per (60). This recursive dependence implies that $\lambda_{t+1}$ relies on itself, making it impossible to solve in closed form.

To find the pair $(x_{t+1}, \lambda_{t+1})$, a binary search procedure is employed. Below, we provide the approach used by Bubeck et al. (2019). Let us denote $\theta = \frac{A_t}{A_{t+1}} \in [0, 1]$. Thus, both $y_t$ and $x_{t+1}$ depend on $\theta$,

$$y_t(\theta) := y_t \overset{(60)}{=} \theta x_t + (1 - \theta)v_t, \quad x_{t+1}(\theta) := x_{t+1} \overset{(59)}{=} \mathcal{N}_{p,M_p}^{\gamma}(y_t(\theta)).$$

Since $\lambda_{t+1} = \frac{a_{t+1}^2}{A_{t+1}}$, we have that $\lambda_{t+1} = \frac{(1-\theta)^2}{\theta} A_t$. Thus, in terms of $\theta$, (58) can be rewritten as

$$\frac{1}{2} \leq \zeta(\theta) \leq \frac{p}{p+1}, \quad \text{where} \quad \zeta(\theta) = \frac{(1-\theta)^2}{\theta} \frac{A_t M_p \cdot \|x_{t+1}(\theta) - y_t(\theta)\|^{p-1}}{(p-1)!}. \tag{61}$$

Note that $\zeta(0) \to +\infty$ and $\zeta(1) = 0$. Hence, one can use binary search to find $\theta$ such that (61) holds true. The complexity of this procedure is $O(\log(1/\varepsilon))$, and a theoretical analysis of binary search procedure can be found in Bubeck et al. (2019). Below we present the total complexity of Algorithm 5.

**Theorem D.3 ((Kamzolov, 2020, Theorem 1))** *For convex function $f$ with $L_p$-Lipschitz-continuous $p$-th derivative, to find $x_T$ such that $f(x_T) - f^* \leq \epsilon$, it suffices to perform no more than $T \geq 1$ iterations of Algorithm 5 with $H_p = \xi L_p$, where $\xi$ and $\gamma$ satisfy $1 \geq 2\gamma + \frac{1}{\xi(p+1)}$, and*

$$T = \tilde{O}\left(\frac{H_p R^{p+1}}{\varepsilon}\right).$$

**Hyperfast Second-order method.** Interestingly, the lower bound for second-order convex optimization, $O\left(\epsilon^{-2/7}\right)$, can be surpassed under higher smoothness assumptions on the objective. Nesterov (2021c) showed that, under the assumption of an $L_3$-Lipschitz third derivative, Algorithm 1 can be implemented using only a second-order oracle, with the third-order derivative approximated via finite gradient differences. This results in a second-order method with $O\left(\epsilon^{-1/4}\right)$ calls to the second-order oracle. The same idea can be applied to Algorithm 1, improving the convergence rate of the second-order method to $\tilde{O}\left(\epsilon^{-1/5}\right)$ Kamzolov (2020).

**Theorem D.4 ((Kamzolov, 2020, Theorem 2))** *For a convex function $f$ with an $L_3$-Lipschitz-continuous third derivative, to find $x_T$ such that $f(x_T) - f^* \leq \epsilon$, it suffices to perform no more than $N_1 \geq 1$ gradient calculations and $N_2 \geq 1$ Hessian calculations in Algorithm 5 with BGDM as the subsolver for the subproblem (59), $H_p = 3L_p/2$, $\gamma = 1/6$, and*

$$N_1 = \tilde{O}\left(\left(\frac{L_3 R^4}{\epsilon}\right)^{\frac{1}{5}} \log\left(\frac{G + H}{\epsilon}\right)\right),$$

$$N_2 = \tilde{O}\left(\left(\frac{L_3 R^4}{\epsilon}\right)^{\frac{1}{5}}\right),$$

*where $G$ and $H$ are the uniform upper bounds for the norms of the gradients and Hessians computed at the points generated by the main algorithm.*

## D.4 PROXIMAL POINT METHOD WITH SEGMENT SEARCH

Another approach for constructing near-optimal tensor methods involves high-order proximal-point type methods Nesterov (2023; 2021a), which are based on the $p$-th-order proximal-point operator:

$$\text{prox}_{p,H}(y) = \underset{x \in \mathbb{E}}{\text{argmin}} \left\{ f_{y,p,H}(x) := f(x) + \frac{H}{p+1}\|x - y\|^{p+1} \right\}. \tag{62}$$

Nesterov (2023) demonstrated that using a single step of a $p$-th-order tensor method to solve (62) results in a convergence rate of $O(\epsilon^{-1/p})$, and moreover, this approach can be accelerated to achieve a rate of $O(\epsilon^{-1/(p+1)})$. Another significant contribution of Nesterov (2023) is the introduction of a proximal-point operator with segment search:

$$\text{Sprox}_{p,H}(y, u) = \underset{x \in \mathbb{E}, \ \tau \in [0,1]}{\text{argmin}} \left\{ f(x) + \frac{H}{p+1}\|x - y - \tau u\|^{p+1} \right\}. \tag{63}$$

Assuming that (63) can be solved exactly, Nesterov (2023) showed that convergence rate of $O(\varepsilon^{-2/(3p+1)})$ can be achieved via different acceleration scheme.

A more practical algorithm was introduced in Nesterov (2021a). Following Nesterov (2023), the authors assumed that the problem (62) can be solved under the following approximate condition:

$$\mathcal{A}_{p,H}^{\gamma}(y) = \{x \in \mathbb{E} : \|\nabla f_{y,p,H}(x)\|_* \leq \beta \|\nabla f(x)\|_*\},$$

where $\gamma \in [0,1)$ is a tolerance parameter. Furthermore, a specific approach for approximating the solution to subproblem (63) was proposed. The resulting method, called the Inexact $p$-th-order Proximal Point Method with Segment Search, is presented in Algorithm 6. Lines 5-14 of Algorithm 6 detail the steps for the approximate solution of (63).

---

**Algorithm 6** Inexact $p$-th-order Proximal Point Method with Segment Search (Nesterov, 2021a, Method (3.6))

---

1: **Input:** $x_0 = v_0$ is starting point, constants $H > 0$, $\gamma \in [0,1)$, total number of iterations $T$, $A_0 = 0$.
2: **for** $t \geq 0$ **do**
3:     Set $u_t = v_t - x_t$.
4:     Compute $x_t^0 \in \mathcal{A}_{p,H}^{\gamma}(x_t)$.
5:     **if** $\langle \nabla f(x_t^0), u_t \rangle \geq 0$, **then**
6:         Define $\phi_t(z) = f(x_t^0) + \langle \nabla f(x_t^0), z - x_t^0 \rangle$, $x_{t+1} = x_t^0$, $g_t = \|\nabla f(x_t^0)\|_*$.
7:     **else**
8:         Compute $x_t^1 \in \mathcal{A}_{p,H}^{\gamma}(v_t)$.
9:         **if** $\langle \nabla f(x_t^1), u_t \rangle \leq 0$, **then**
10:           Define $\phi_t(z) = f(x_t^1) + \langle \nabla f(x_t^1), z - x_t^1 \rangle$, $x_{t+1} = x_t^1$, $g_t = \|\nabla f(x_t^1)\|_*$.
11:         **else**
12:           Find values $0 \leq \tau_t^1 \leq \tau_t^2 \leq 1$ with points $w_t^1 \in \mathcal{A}_{p,H}^{\gamma}(x_t + \tau_t^1 u_t)$ and $w_t^2 \in \mathcal{A}_{p,H}^{\gamma}(x_t + \tau_t^2 u_t)$ satisfying

$$\beta_t^1 \leq 0 \leq \beta_t^2, \quad \text{and} \quad \alpha_t(\tau_t^1 - \tau_t^2)\beta_t^1 \leq \tfrac{1}{2} \left[\tfrac{1-\gamma}{H}\right]^{1/p} g_t^{\frac{p+1}{p}},$$

where $\beta_t^1 = \langle \nabla f(w_t^1), u_t \rangle$, $\beta_t^2 = \langle \nabla f(w_t^2), u_t \rangle$, $\alpha_t = \frac{\beta_t^2}{\beta_t^2 - \beta_t^1} \in [0,1]$, and

$$g_t = \left[\alpha_t \|\nabla f(w_t^1)\|_*^{\frac{p+1}{p}} + (1 - \alpha_t)\|\nabla f(w_t^2)\|_*^{\frac{p+1}{p}}\right]^{\frac{p}{p+1}}.$$

          Set

$$\phi_t(z) = \alpha_t \left(f(w_t^1) + \langle \nabla f(w_t^1), z - w_t^1 \rangle\right) + (1 - \alpha_t)\left(f(w_t^2) + \langle \nabla f(w_t^2), z - w_t^2 \rangle\right),$$
$$x_{t+1} = \alpha_t w_t^1 + (1 - \alpha_t)w_t^2.$$

13:         **end if**
14:     **end if**
15:     Compute $a_{t+1} > 0$ from equation $\frac{a_{t+1}^2}{A_t + a_{t+1}} = \frac{1}{2}\left[\frac{1-\gamma}{H}\right]^{1/p} g_t^{\frac{1-p}{p}}$
16:     Set $A_{t+1} = A_t + a_{t+1}$ and update $\psi_{t+1}(z) = \psi_t(z) + a_{t+1}\phi_t(z)$
17:     Set $v_{t+1} = \operatorname*{argmin}_{z \in \mathbb{E}} \psi_{t+1}(z)$
18: **end for**
19: **return** $x_T$

---

**Theorem D.5 ((Nesterov, 2021a, Theorem 2))** *For smooth convex function $f$ to find $x_T$ such that $f(x_T) - f^* \leq \epsilon$, it suffices to perform no more than $T \geq 1$ iterations of Algorithm 6, where*

$$T = O\left(\left[\frac{HR^{p+1}}{\varepsilon}\right]^{\frac{2}{3p+1}}\right).$$

Line 12 requires additional bisection search with complexity of $O\left(\frac{HD^{p+1}}{\varepsilon}\right)$ (Nesterov, 2021a, Theorem 4). This results in the following upper bound for the number of evaluations of $w \in \mathcal{A}_{p,H}^{\gamma}(x)$ during the execution of Algorithm 6 $O\left(\left[\frac{HD^{p+1}}{\varepsilon}\right]^{\frac{2}{3p+1}} \log \frac{HD^{p+1}}{\varepsilon}\right)$.

Under the additional assumption of an $L_p$-Lipschitz continuous $p$-th derivative of $f$, the inclusion $w \in \mathcal{A}_{p,H}^{\gamma}(x)$ can be achieved by performing one inexact tensor step with specific choice of parameters $\beta$ and $M_p$: $w \in \mathcal{N}_{p,M_p}^{\beta}(x)$ (Nesterov, 2023, Section 3) (Nesterov, 2021a, Section 5.1). This makes Algorithm 6 a near-optimal tensor method, comparable to Gasnikov et al. (2019b); Bubeck et al. (2019); Jiang et al. (2019). However, it differs in nature: while the latter methods are based on A-NPE-type approaches, Algorithm 6 follows an interior-point-type framework.

For the case when $p = 3$, the tensor step can be efficiently performed using BDGM in $O\left(\log 1/\varepsilon\right)$ iterations. As demonstrated in Nesterov (2021c); Kamzolov (2020), a second-order implementation of a third-order tensor method can be achieved by approximating the third-order derivative using finite gradient differences. However, in practice, this approximation may suffer from numerical instability. For Algorithm 6 another approach is available: the interior-point subproblem (62) can be solved using a second-order method Nesterov (2021a), which provides a more reliable alternative to finite gradient differences. Under the assumption of an $L_3$-Lipschitz continuous third derivative of $f$, Algorithm 6 achieves convergence $\tilde{O}\left(\varepsilon^{-1/5}\right)$.

### D.5 Optimal Tensor Method

An Optimal Tensor Method was recently proposed by Kovalev & Gasnikov (2022); Carmon et al. (2022), improving upon the convergence of near-optimal tensor methods Gasnikov et al. (2019a); Bubeck et al. (2019); Jiang et al. (2019). The convergence rate was enhanced from $O\left(\varepsilon^{-2/(3p+1)} \log(1/\varepsilon)\right)$ to $O\left(\varepsilon^{-2/(3p+1)}\right)$, matching the lower bound $\Omega\left(\varepsilon^{-2/(3p+1)}\right)$ Arjevani et al. (2019). Similar to near-optimal methods, the Optimal Tensor Method is based on the A-HPE framework proposed by Monteiro & Svaiter (2013).

Before describing the Optimal Tensor Method, we introduce some necessary notations. Let $\Phi_p^g$ denote the $p$-th order Taylor approximation of the function $g$:

$$\Phi_p^g(x, y) = g(y) + \sum_{k=1}^{p} \frac{1}{k!} D^k g(y)[x - y]^k. \tag{64}$$

Additionally, note that $\Phi_p^f(x, y) = \Phi_p(x, y)$ as defined in (8). We also define the function $g_\lambda(x, y) = f(x) + \frac{1}{2\lambda}\|x - y\|^2$.

The main distinction from near-optimal methods lies in the procedure used to find the pair $(x_{t+1}, \lambda_{t+1})$. Instead of first computing $x_{t+1}$ and then using a binary search to determine $\lambda_{t+1}$, as done in previous approaches, Kovalev & Gasnikov (2022) first select the parameter $\lambda_{t+1}$ and then compute $x_{t+1}$. This procedure, known as the Tensor Extragradient Method, is shown in lines 6- 10 of Algorithm 7. This method converges in a constant number of iterations, leading to the optimal convergence rate of $O\left(\varepsilon^{-2/(3p+1)}\right)$ for Algorithm 7.

**Theorem D.6 ((Kovalev & Gasnikov, 2022, Theorem 5))** *Let $M_p = L_p$ and $\sigma = 1/2$. Let*

$$\nu = \left(\frac{(3p+1)^p C_p(M_p, \sigma) R^{p-1}}{2^p \sqrt{p}} \cdot \left(\frac{1+\sigma}{1-\sigma}\right)^{\frac{p-1}{2}}\right)^{-1},$$

$$where \quad C_p(M_p, \sigma) = \frac{p^p M_p^p (1 + \sigma^{-1})}{p!(pM_p - L_p)^{p/2}(pM_p + L_p)^{p/2-1}}.$$

*Then, for convex function $f$ with $L_p$-Lipschitz-continuous $p$-th derivative, to find $x_T$ such that $f(x_T) - f^* \leq \epsilon$, it suffices to perform no more than $T \geq 1$ iterations of Algorithm 7, where*

$$T = 5D_p \cdot \left(L_p R^{p+1}/\epsilon\right)^{\frac{2}{3p+1}} + 7,$$

---

**Algorithm 7** Optimal Tensor Method (Kovalev & Gasnikov, 2022, Algorithm 4)

---

1: **Input:** $x_0 = v_0$ is starting point, constants $M_p$, $\sigma \in (0,1)$, total number of iterations $T$, $A_0 = 0$, sequence $a_t = \nu t^{(3p-1)/2}$ for some $\nu > 0$.
2: **for** $t \geq 0$ **do**
3:    $A_{t+1} = A_t + a_{t+1}$, $\lambda_{t+1} = \frac{a_{t+1}^2}{A_{t+1}}$
4:    $y_t = \frac{A_t}{A_{t+1}}x_t + \frac{a_{t+1}}{A_{t+1}}v_t$
5:    $y_t^0 = y_t$, $k = 0$
6:    **repeat**
7:      $x_t^k = \underset{y \in \mathbb{E}}{\operatorname{argmin}} \left\{ \Phi_p^{g_{\lambda_t}(\cdot,y_t)}(y, y_t^k) + \frac{pM_p}{(p+1)!}\|y - y_t^k\|^{p+1} \right\}$
8:      $y_t^{k+1} = y_t^k - \left( \frac{M_p\|x_t^k - y_t^k\|^{p-1}}{(p-1)!} \right)^{-1} \nabla g_{\lambda_{t+1}}(x_t^k, y_t)$
9:      $k = k + 1$
10:    **until** $\|\nabla g_{\lambda_{t+1}}(x_t^k, y_t)\| \leq \sigma\lambda_t^{-1}\|x_t^k - y_t\|$
11:    $x_{t+1} = x_t^{k-1}$
12:    Update $v_{t+1} = v_t - a_{t+1}\nabla f(x_{t+1})$
13: **end for**
14: **return** $x_T$

---

with $D_p$ is defined as follows:

$$D_p = \left( \frac{3^{\frac{p+1}{2}}(3p+1)^{p+1}p^p(p+1)}{2^{p+2}\sqrt{p}p!(p^2-1)^{\frac{p}{2}}} \right)^{\frac{2}{3p+1}}.$$

# E   EXPERIMENTAL DETAILS

**Setup.** All methods and experiments were performed using Python 3.11, PyTorch 2.2.2, on a 13-inch MacBook Pro 2019 with 1,4 GHz Quad-Core Intel Core i5 and 8GB memory. All computations are done in torch.double. All methods are implemented as PyTorch 2 optimizers.

**Logistic Regression.** The logistic regression problem can be formulated as

$$f(x) = \frac{1}{n}\sum_{i=1}^n \log\left(1 + e^{-b_i\langle a_i,x\rangle}\right) + \frac{\mu}{2}\|x\|_2^2, \tag{65}$$

where $a_i \in \mathbb{R}^d$ are data features and $b_i \in \{-1; 1\}$ are data labels for $i = 1, \ldots, n$.

We present results on the a9a dataset ($d = 123, n = 32561$) and w8a ($d = 300, n = 49749$) from LibSVM by Chang & Lin (2011). We choose the starting point $x_0 = 3e$, where $e$ is a vector of all ones. This choice of $x_0$ allows us to show the convergence of the methods from a far point. For Figures 6, 5 and 7a, we choose the regularizer $\mu = 10^{-4}$ to get strongly-convex function $f$. For Figures 2,4, and 8, we choose the regularizer $\mu = 0$ to get a convex function $f$. For the better conditioning, we normalize data features $\|a_i\| = 1$. For the normalized case, we choose theoretical $L_2 = 0.1$. We set $L_3 = L_2 = 0.1$ to demonstrate the convergence rates for the same constants $L$. Note, that actual $L_3$ is smaller than 0.1.

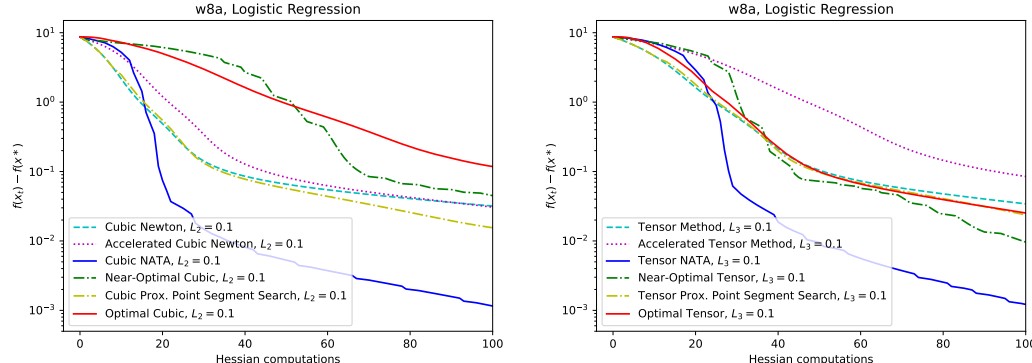

Figure 8: Comparison of different cubic and tensor acceleration methods on Logistic Regression for w8a dataset from the starting point $x_0 = 3e$, where $e$ is a vector of all ones.

**Third-order Nesterov's lower-bound function.** The $l_2$-regularized third order Nesterov's lower-bound function from Nesterov (2021b) has the next form

$$f(x) = \frac{1}{4}\sum_{i=1}^{d-1}(x_i - x_{i+1})^4 - x_1 + \frac{\mu}{2}\|x\|_2^2. \tag{66}$$

For Figures 1 and 7b, we set $d = 20$, $\mu = 10^{-3}$, we've tuned $L_3 = L_2 = 10$..

**Poisson regression.** Poisson regression is a type of generalized linear model used for analyzing count data and contingency tables. It assumes that the response variable $b_i$ follows a Poisson distribution, and the logarithm of its expected value can be expressed as a linear combination of unknown parameters. The Poisson regression function has the next form

$$f(x) = \sum_{i=1}^{n} e^{\langle a_i, x \rangle} - b_i \langle a_i, x \rangle, \tag{67}$$

where $a_i \in \mathbb{R}^d$ are data features and $b_i \in \{0, 1, \ldots, k, \ldots\}$ are countable targets.

We present results for synthetic data: $d = 21$, $n = 6000$. We set $L_1 = L_2 = L_3 = 1$ and $x_0 = e$ is all ones.

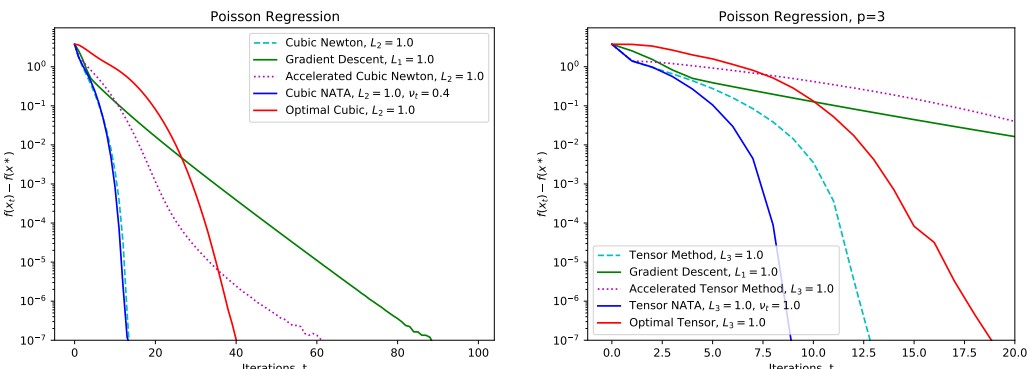

Figure 9: Comparison of different cubic and tensor accelerated methods on Poisson Regression.

The Cubic Regularized Newton (CRN) method and NATA with a tuned parameter $\nu$ demonstrate the best performance in Figure 9 (Left). Notably, CRN exhibits rapid superlinear convergence, likely due to the strong convexity properties of the loss function. Interestingly, NATA with the tuned $\nu$ manages to match CRN's convergence rate. While Optimal Acceleration is slower than both CRN and NATA, it also achieves global superlinear convergence. In Figure 9 (Right) for $p = 3$, the Tensor Nata method is the fastest, followed by the Basic Tensor Method, with the Optimal Tensor method ranking third. All three methods exhibit global superlinear convergence. The classical Nesterov

Tensor Acceleration method is the slowest, likely due to its small default $\nu$. Notably, the tensor-based methods outperform their cubic counterparts.

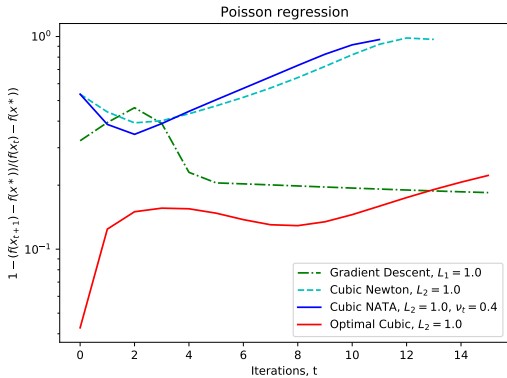

Figure 10: Comparison of the methods by the relative value $1 - \frac{f(x_{t+1})-f^*}{f(x_t)-f^*}$.

The global superlinear performance of these accelerated second-order methods in Figure 10 raises the hope of establishing theoretical results on global superlinear convergence for accelerated second-order methods.

