# OpenReview forum: "OPTAMI: Global Superlinear Convergence of High-order Methods"
_ICLR.cc/2025/Conference — ICLR 2025 Poster_

### Official Review · Reviewer_TJTR · 2024-10-27

**Soundness:** 3
**Presentation:** 2
**Contribution:** 2
**Rating:** 5
**Confidence:** 4

**Summary:**

The paper presents two main results: the first shows that the basic high-order methods (the Cubic Regularized Newton Method and Basic Tensor Method) achieve global superlinear convergence for μ-strongly star-convex functions.
Second, it proposes a technique for choosing the parameters of an existing method (Nesterov Accelerated Tensor Method), evaluates its convergence speed, and shows its usefulness in numerical experiments.
Furthermore, the paper introduces an open-source computational library called OPTAMI for high-order methods.

**Strengths:**

It is great to show that cubic Newton and Basic Tensor methods can achieve superlinear convergence for strongly convex functions.
This paper also includes the necessary equations and propositions for following the proof of superlinear convergence in the main text, and explains the proof strategy leading to the most important theorem, Theorem 3.4, in an easy-to-understand manner.

 I also think that providing an open-source computational library for high-order methods would be very helpful to subsequent researchers.

**Weaknesses:**

[Structure of the paper]
Section 3 and Section 4 are not well connected, and the paper seems to contain two disparate contents; it would be better to divide the paper into two papers, and strengthen the contents of each, for example, in the following ways. The algorithms and classes of functions are different between Sections 3 and 4.

For example, for the contents of Section 3, if the cost of solving the subproblem in each iteration can be evaluated, what is the total computational complexity? How attractive is it compared to the total computational complexities of other algorithms? Even if the number of iterations (i.e., the worst-case iteration complexity) is reduced, on the other hand, if the computational complexity at each iteration explodes, it will not be attractive as an algorithm, so it would be better to provide the total computational complexity. If the subproblem is to be solved iteratively, can't we allow it to be solved inexactly and include the error in the iteration complexity and total computational cost?

For example, for the contents of Section 4, if we consider strongly star-convex, how can the theoretical guarantee given in Theorem 4.1 be changed? As future work, the authors wrote this type of question, but what trends do you see, at least in numerical experiments? More precisely, what would Figure 3(b) look like in the case of strongly convex (21) with positive $\mu$ (perhaps the horizontal axis might need to be the iteration number)?

----------

[Various unclear descriptions]
There are unclear descriptions and various minor errors, giving the impression that this paper was written in haste.

For example, lines 62-74 introduce the existing studies and describe their global convergence rates, but the assumption of the function $f$ is not clearly stated. Before that, there is a definition of star-convex, etc., but I do not think that those existing studies assume star-convex. This is because the Hessian matrix in (7) is not necessarily positive definite when the function is nonconvex.
The others are line 80: upperbound --> upper bound, NPE in line 109 should be written as Newton Proximal Extragradient (NPE), in line 123 the methods performs --> the method performs, etc. Line 193 defines the norm $||x||$ using a matrix B, but which kind of matrices are specifically used in your claims? If we always assume $B=I$, why use $B$ to define the norm? I won't point out any more, but I would like the authors to review the paper again and and reduce this type of typo.

----------

[Insufficient reference]
The survey of existing studies does not seem to be sufficient. For example, in line 97, the authors write “poor global convergence” for the quasi-Newton method, but I do not think they are aware of the following paper. The paper also shows “Global Non-Asymptotic Superlinear Convergence” as well as this paper.

Qiujiang Jin, Ruichen Jiang, Aryan Mokhtari, "Non-asymptotic Global Convergence Rates of BFGS with Exact Line Search",
arXiv:2404.01267, 2024.

 Although the authors mention the Jiang, Jin, and Mokhtari paper (COLT 2023) in Appendix A, I think the global convergence rates of quasi-Newton methods should be mentioned around lines 97-98, including the above-mentioned papers. Can it no longer be called “poor global convergence”?

**Questions:**

1) What are the total computational complexities for the Cubic Regularized Newton Method and Basic Tensor Method if the costs of solving the subproblems in each iteration can be evaluated? How attractive are they compared to the total computational complexities of other algorithms?

2) If the subproblems of the Cubic Regularized Newton Method and Basic Tensor Method are to be solved iteratively, can't we allow them to be solved inexactly and include the errors in total computational costs?

3) About the proposed algorithm, Newterov Accelerated Tensor Method with At-Adaptation (NATA). Can the authors show anything about the iteration complexities for the strongly (star-)convex case? If it is difficult, show us the best iteration complexity for the classical Nesterov accelerated tensor method when the function $f$ is strongly convex, and tell us why it is difficult to derive the complexity for NATA.

4) For the strongly (star-)convex case, how does NATA perform in numerical experiments? More precisely, what would Figure 3(b) look like in the case of strongly convex (21) with positive $\mu$ (perhaps the horizontal axis might need to be the iteration number)?

---

> ### Author Response · Authors · 2024-11-21
>
> Dear Reviewer TJTR,
>
> Thank you for your valuable feedback! We appreciate your recognition of the strengths of our work, including the established superlinear convergence of second-order methods and our open-source library. We also value your constructive suggestions for areas requiring further discussion and improvement.
> Below, we address your questions. We have also uploaded a revised version of the paper and kindly invite you to review it.
> ___
>
> ### Weakness 1. Structure of the paper
> We realize that our initial presentation may have created the impression that NATA and the superlinear convergence results are separate topics. However, these two components are closely linked, as both address open challenges we faced during the development of the OPTAMI library.
> For further details, we invite you to review the common commentary, "Modified Structure of the Work," where we explain the narrative behind our approach and the relationship between these elements. The paper has been updated to reflect these clarifications.
>
> ### Weakness 2. Questions 1 and 2. Subproblem.
>
> Following the classical literature, we prove theoretical convergence results under the assumption of exact computations and exact subproblem solution. Introducing inexactness at this stage could overcomplicate the paper with technical details that might obscure the main ideas and results. We believe that analyzing various types of inexactness in the method could form the basis of a separate, technically focused paper more suited for an optimization journal [1],[2].
> From a practical point of view, subproblems can often be solved inexactly, for example, Basic Tensor Method implemented in OPTAMI with inexact iterative subsolver. Our experiments demonstrate that this inexactness does not compromise global superlinear convergence. Efficiently solving cubic regularization and tensor subproblems remains an active area of research, with numerous dedicated papers [3], [4], [5], [6]. For this reason, we have chosen to leave this topic outside the scope of our current work.
>
> [1] Nesterov, Yurii. "Inexact basic tensor methods for some classes of convex optimization problems." Optimization Methods and Software 37.3 (2022): 878-906.
>
> [2] Grapiglia, Geovani Nunes, and Yu Nesterov. "On inexact solution of auxiliary problems in tensor methods for convex optimization." Optimization Methods and Software 36.1 (2021): 145-170.
>
> [3] Carmon, Yair, and John Duchi. "Gradient descent finds the cubic-regularized nonconvex Newton step." SIAM Journal on Optimization 29.3 (2019): 2146-2178.
>
> [4] Thomsen, Daniel Berg, and Nikita Doikov. "Complexity of Minimizing Regularized Convex Quadratic Functions." arXiv preprint arXiv:2404.17543 (2024).
>
> [5] Jiang, Rujun, Man-Chung Yue, and Zhishuo Zhou. "An accelerated first-order method with complexity analysis for solving cubic regularization subproblems." Computational Optimization and Applications 79 (2021): 471-506.
>
> [6] Gao, Yihang, Man-Chung Yue, and Michael Ng. "Approximate secular equations for the cubic regularization subproblem." Advances in Neural Information Processing Systems 35 (2022): 14250-14260.
>
> ### Weakness 3. Question 3 and 4. Strong star-convexity
> Thank you for the insightful questions and suggestions. In response, we have added experiments for various accelerated methods on strongly convex functions, as shown in Figure 5. Interestingly, most of these methods appear to exhibit global superlinear convergence without any specific theoretical adaptation for strongly convex functions, which is quite surprising.
>
> Regarding the proofs, we can derive classical theoretical results for the restarted version of NATA, similar to existing methods. However, we currently do not have a way to prove global superlinear convergence for either classical Nesterov Acceleration or NATA. As such, we have highlighted this as an open problem and a potential direction for future research in the Conclusion section.
>
> ### Weakness 4. Various unclear descriptions
>
> Thank you for highlighting the unclear descriptions and minor errors. We have addressed these issues during the reorganization of the paper. The specific changes are as follows:
> 1) We now begin the paper by focusing on convex functions, ensuring that the rates presented in the introduction are valid.
> 2) We have corrected the “upperbound” and removed NPE.  Matrix $B$ and corresponding norm have been introduced to align with the more general notation used by Yurii Nesterov in his papers [7], [1], [2]. However, if you believe this adds unnecessary complexity, we are open to removing it.
>
> [7] Nesterov, Yurii. "Implementable tensor methods in unconstrained convex optimization." Mathematical Programming 186 (2021): 157-183.

---

> ### Author Response · Authors · 2024-11-21
>
> ### Weakness 5. Insufficient reference.
> As part of the reorganization, we entirely removed the discussion regarding the “poor global convergence” rate of the quasi-Newton method. Instead, we introduced a concise paragraph on Hessian approximations to motivate the exploration of second-order methods. Additionally, we incorporated more citations to strengthen the section on Quasi-Newton methods and Hessian approximations.

---

> > ### Comment · Reviewer_TJTR · 2024-11-24
> > **Thanks for the response.**
> >
> > Thank you to the authors for their responses. In this revision, the structure of the paper has been improved, making it much easier to read than the previous version. Your time and effort are appreciated.
> >  However, my concern still remains because the structure change does not change the content. The NATA proposed in Section 3 does not appear in Section 4, which analyses the existing methods: the Cubic Newton method and the Basic Tensor method. I understand there is no way to fix this point.

---

> > > ### Author Response · Authors · 2024-11-27
> > >
> > > Dear Reviewer TJTR,
> > >
> > > Thank you for acknowledging the improvements in structure and readability in the revised version of our paper. We greatly appreciate your thoughtful feedback.
> > >
> > > Could you kindly clarify your current main concern to ensure there is no misunderstanding? At the moment, it is confusing and unclear to us.
> > >
> > >   Best regards, Authors.

---

> > > > ### Comment · Reviewer_TJTR · 2024-12-01
> > > > **Sorry for my unclear comments**
> > > >
> > > > I'm still concerned about the total computational complexities of the Cubic Regularized Newton Method and Basic Tensor Method. Even if the number of iterations (i.e., the worst-case iteration complexity) is reduced, on the other hand, if the computational complexity at each iteration explodes, it will not be attractive as an algorithm.
> > > >  I understood your answer to be saying that it is difficult to do so by citing several existing studies. On the contrary, are there any studies that evaluate the total computational complexity with cubic regularized Newton or higher-order algorithms like yours?

---

> ### Author Response · Authors · 2024-12-04
>
> Dear Reviewer TJTR,
>
> Thank you for the clarification!
>
> Allow us to explain our perspective on this matter. Our theoretical results are derived under the standard assumptions of exact computations and exact subproblem solutions. It is a common and acknowledged practice in optimization community. To support this perspective, we highlight several influential optimization papers from well-known research groups accepted at major conferences or journals which assume exact computations and exact subproblem solution [1, 2, 3, 4, 5].  We believe that theoretical improvements within this exact regime is both normal and valuable, and it should not be considered a reason for rejection.
>
> From the practical perspective, our Cubic Regularised Newton is implemented as it was introduced in the original paper with the same computational complexity. So, there is no overhead computation to get the provided experiments. As we mentioned in limitations, all second-order methods with exact Hessian “have computational and memory limitations in high-dimensional spaces, due to the need for Hessian calculations. There are, however, approaches to overcome this, such as using first-order subsolvers or inexact Hessian approximations like Quasi-Newton approximations (BFGS, L-SR1). In this paper, we focus on the exact Hessian to analyze methods’ peak performance.” We view our results as fundamental discovery for second-order methods which explains their practical properties. Next, in future research, they can be smartly coupled with inexact Hessian approximations or first-order subsolvers to make them dominant in both low and high-dimension settings.
>
> We hope this explanation clarifies our perspective and addresses your concerns.
>
> Best regards,
> Authors
>
> [1] Doikov, N., Mishchenko, K. and Nesterov, Y., 2024. Super-universal regularized newton method. SIAM Journal on Optimization, 34(1), pp.27-56.
>
> [2] Bubeck, S., Jiang, Q., Lee, Y.T., Li, Y. and Sidford, A., 2019, June. Near-optimal method for highly smooth convex optimization. In Conference on Learning Theory (pp. 492-507). PMLR.
>
> [3] Antonakopoulos, K., Kavis, A. and Cevher, V., 2022. Extra-newton: A first approach to noise-adaptive accelerated second-order methods. Advances in Neural Information Processing Systems, 35, pp.29859-29872.
>
> [4] Gower, R., Goldfarb, D. and Richtárik, P., 2016, June. Stochastic block BFGS: Squeezing more curvature out of data. In International Conference on Machine Learning (pp. 1869-1878). PMLR
>
> [5] Jiang, R., Raman, P., Sabach, S., Mokhtari, A., Hong, M. and Cevher, V., 2024, April. Krylov Cubic Regularized Newton: A Subspace Second-Order Method with Dimension-Free Convergence Rate. In International Conference on Artificial Intelligence and Statistics (pp. 4411-4419). PMLR.

---

### Official Review · Reviewer_Pnwc · 2024-11-02

**Soundness:** 3
**Presentation:** 3
**Contribution:** 2
**Rating:** 6
**Confidence:** 3

**Summary:**

In this work, the authors studied the convergence rate of high-order methods, e.g., the cubic regularized Newton method and the basic tensor method. The authors proved the global superlinear convergence of both methods for $\mu$-strongly star-convex functions and $\mu_q$-uniformly star-convex functions, respectively. In addition, a variant of accelerated high-order method, named NATA, was proposed and compared with other accelerated high-order methods.

**Strengths:**

The paper is well-organized and easy to follow. The results are novel and should be interesting to the audience from machine learning and optimization fields. The problems studied in this work are important and applicable to certain practical problems where the computational time is not a critical constraint. The proof in the main manuscript should be correct, while I do not have time to check the proofs in the appendix due to the time limit.

**Weaknesses:**

The main problem with the paper is that the first part (superlinear convergence rate of high-order methods) and the second part (NATA algorithm) seem to be independent and can be separated into two papers. I feel that these two parts considered two different topics. The first one is mostly theoretical and is about non-accelerated methods, while the second one is about accelerated methods and their empirical performance. I would suggest the authors split the paper into two and include more details to the content. For example, the intuition behind the design of NATA.

In addition, I think the current introduction section is too lengthy. Considering the page limit of the conference, the background knowledge can be simplified and moved to the appendix, since it can be easily found in textbooks and literature.

Finally, I think the authors could include more details to the experiments. For example, in Figure 3, I wonder if the Tensor NATA converges faster with a carefully chosen $v_t$? This is not discussed. Also, it would be better if the authors could provide more intuition behind the current design of searching $v_t$ in sub-iterations instead of fixing $v_t$ to be a constant. Since the performance of a fixed $v_t$ is better than that of a adaptive $v_t$, I wonder if this design is unnecessary. Furthermore, I think it will be helpful if the authors could provide the running time comparison. This is because the solving time of the sub-problem in each iteration may be non-negligible and comparable to the computation time of the Hessian matrix. Especially, for the NATA method, the sub-problem needs to be solved several times in each iteration.

**Questions:**

I have a few other minor comments for the authors to consider:

- Line 143: I think it should be $\epsilon > c_3 r$?

- Line 295: "where the sublinear rate outperforms the linear rate" is a little confusing. Maybe the authors meant the high-order methods have not entered the linear convergence region and the convergence rate is sublinear at the beginning?

- For the CRN method, did the authors prove the superlinear convergence for $\mu_q$-uniformly star-convex functions? If so, it may be better to state the results in Theorem 3.2. Currently, I cannot find the results for $\mu_q$-uniformly star-convex functions and the CRN method.

---

> ### Author Response · Authors · 2024-11-21
>
> Dear Reviewer Pnwc,
> Thank you for your valuable feedback. We are grateful for your recognition of the strengths of our work and for your thoughtful and constructive suggestions, including the presentation style and the perceived logical disconnect between the two parts of our work in the initial version.
> We have uploaded a revised version in which we have tried to address your suggestions. We kindly ask you to review the updated manuscript along with the common commentary "Modified Structure of the Work."
>
> ___
>
> ### Major comments:
> - We regret that our initial presentation style gave the impression that NATA and the superlinear convergence rate are two independent topics. NATA and the theoretical proof of superlinear convergence address open problems we encountered during the development of the OPTAMI library. For more details, please review the common commentary "Modified Structure of the Work", where we outline the narrative behind our approach and the connection between these components. We have updated the paper accordingly.
>
> - We refined the Introduction section by moving technical details to a specific subsection in Section 2, which focuses on the contents of the library. The paper has been updated accordingly.
>
> -  As you recommended, we have added NATA with tuned $\nu^t$ for every figure in the main paper. We will also include them in the experiments presented in the Appendix later. The main difference between the tuned and adaptive versions lies in the number of main parameters: adaptive NATA has one primary parameter, $L$, while NATA with tuned $\nu^t$ involves two parameters, $L$ and $\nu$. The additional parameter is a limitation of the tuned method compared to other acceleration techniques, which typically require only one main parameter.
> In the paper, we emphasize that the method with tuned $\nu$ generally outperforms the adaptive version in terms of efficiency, as it avoids the need for additional computations associated with adaptive search. However, this advantage depends on proper tuning of $\nu^t$ — without accurate tuning, the method may diverge. From our perspective, both adaptive and tuned NATA have their applications. If the adaptive search is available, the adaptive NATA  is useful as has less parameters. On the other hand, while the tuned version can significantly accelerate the method (compared to classical Nesterov acceleration) in scenarios where exact function values are not accessible.
>
> - We believe that using "Hessian computations" as the comparison axis is the fairest metric. All cubic methods solve the same type of subproblem and require nearly the same computational effort per iteration. Specifically, one Hessian computation corresponds to solving a single cubic subproblem. Furthermore, like adaptive NATA, other near-optimal and optimal acceleration methods perform line searches and may compute multiple Hessians per iteration.  For example, Prox Point Segment Search methods typically involve around three Hessian computations per iteration. This consistency makes "Hessian computations" a fair and reliable axis for comparing acceleration methods.
>
> ### Questions
> 1. Thank you. We fixed the misprint.
>
> 2. Thank you. We rephrased this line. Please let us know, if it is more clear now.
>
> 3. No, the CRN converges globally superlinearly only for strongly star-convex functions, not for uniformly star-convex functions. On the other hand, the $p$-th order Basic Tensor method achieves global superlinear convergence for $\mu_q$-uniformly star-convex functions when $p \geq q-1$.

---

### Official Review · Reviewer_WM8N · 2024-11-02

**Soundness:** 4
**Presentation:** 4
**Contribution:** 3
**Rating:** 8
**Confidence:** 4

**Summary:**

This paper first establishes the global superlinear convergence rate for second-order methods in optimizing strongly star-convex functions, and its higher-order extension for optimizing uniformly star-convex functions, then proposes a variant of Nesterov accelerated tensor method which demonstrates superior performance in numerical experiments consistent with the theoretical faster convergence rate, and finally presents a systematical numerical comparison across mainstream second-order methods.

**Strengths:**

1. The proposed acceleration variant of tensor method achieves better empirical performance than the existing (near)-optimal accelerated second-order methods.
2. All methods are systematically implemented and released as a library.

**Weaknesses:**

1. The global linear rate is established by relaxing the required accuracy to exceed the radius of the quadratic convergence region.
2. Some typos:
- line 143 "$\epsilon \leq c_3 r$" --> $\epsilon > c_3 r$
- Eq (20) $t \rightarrow 0$ --> $t \rightarrow \infty$?

**Questions:**

1. How do the authors interpret this trade-off between accuracy and rate of convergence? If the requirement on accuracy is relaxed and the conditions are different, how can the superlinear convergence rate be seen as an improvement compared to previous results?
2. In addition to the first question, Song et. al. 2021 proposed an acceleration framework that matches the lower bound established by Arjevani et al. 2019. Shouldn't that be seen as the optimal rate for this setting of optimizing strongly convex functions with second-order methods?

Chaobing Song, Yong Jiang, and Yi Ma. Unified acceleration of high-order algorithms under general holder
continuity. SIAM Journal on Optimization, 31(3):1797–1826, 2021.

---

> ### Author Response · Authors · 2024-11-21
>
> Dear Reviewer WM8N,
> We sincerely appreciate your comprehensive and insightful review of our paper and your recognition of the strengths of our work. Below, we try to address your questions and comments.
>
> ___
> ### Weakness 1. Questions 1 and 2
>
> The existing results about the convergence rate of second-order methods for strongly-convex functions primarily address the regime where $\varepsilon < \frac{\mu^3}{L_2^2}$, as seen in both the lower-bounds (Theorem 1, Arjevani et al. 2019) and upper-bounds (Formula 1.14, Song et al. 2021). For instance, in Formula 1.14 from Song et al. 2021, if $\varepsilon >  \frac{\mu^3}{L_2^2}$, the term $\log( \log(\frac{\mu^3}{L_2^2 \varepsilon}))$ becomes undefined. This implies that the method from Song et al. 2021 is optimal for the setting of optimizing strongly convex functions with second-order methods when $\varepsilon < \frac{\mu^3}{L_2^2}$.
>
> In our work, we focus on the less-explored regime where $\varepsilon > \frac{\mu^3}{L_2^2}$, which corresponds to scenarios where a solution can be less precise. To visualize this regime better, consider a function $f(x)$ with $L_2=1$ and $\mu=10^{-4}$. In this case, Formula 1.14 imposes $\varepsilon < 10^{-12}$, a value far smaller than practical needs. In some applications, achieving an approximate solution with accuracy around $\varepsilon = 10^{-6}$ may be sufficient, which is the focus of our results. Furthermore, once the method with a global superlinear rate enters the quadratic convergence region, it transitions to classical convergence rates. Thus, our results complement the existing literature by addressing a different regime rather than introducing a trade-off. Additionally, our results provide valuable insights into the practical performance of second-order methods. In summary, we address the less-explored setting where $\varepsilon > \frac{\mu^3}{L_2^2}$, complementing and extending the existing literature rather than contradicting it. We have also corrected all the typos you highlighted. Thank you for pointing them out!

---

> > ### Comment · Reviewer_WM8N · 2024-12-03
> >
> > Thank the authors for the reply. I'm willing to keep the score.

---

### Official Review · Reviewer_BoHs · 2024-11-04

**Soundness:** 3
**Presentation:** 3
**Contribution:** 3
**Rating:** 6
**Confidence:** 4

**Summary:**

The paper focuses on the analysis of high order methods for nearly convex functions (i.e. star-convex or convex) with additional growth properties. The authors leverage a strong star-convexity and a uniform star-convexity assumption to prove the global superlinear convergence of the Cubic Regularized Newton Method and the Basic Tensor Method. In addition, they introduce an adaptive variant of Nesterov Accelerated Tensor Method called NATA (for Nesterov Accelerated Tensor Method with $A_t$-Adaptation) which solves the problem of having too conservative parameters. Theoretical convergence guarantees are given as well as numerical experiments highlighting good performance. The authors also provide OPTAMI, a python library for high order methods.

**Strengths:**

Overall, I think that the contributions of the paper are valuable and the technical content would be sufficient for publishing the paper in ICLR. Moreover, I did not find any major mistake in the proofs. The library seems to be qualitative and the numerical experiments are satisfactory to me.

**Weaknesses:**

I am not convinced by the structure of the paper and the way it is written. It seems to me that the authors try to answer too many questions for a 10 pages paper. Also, I think that the theoretical claims are not discussed enough. Detailed comments can be found below. The main problem I have with the current version of the paper is that it lacks a clear unified story and it seems  an aggregation of results. In my opinion, this could be improved by modifying the structure of the paper.

The OPTAMI library is only mentioned but never properly introduced. I think that at least a paragraph should be dedicated to it or else, that it should be removed from the title.
About the structure:

   1) I believe that the introduction is already too technical although it is well explained. It also states too many problems and, due to a lack of structure, it can confuse the reader. This is highlighted by the fact that there are three questions at the end of the introduction. My suggestion is to keep the detailed discussions for later sections and do a way shorter and high-level introduction. Also, I would try to use as few equations as possible.

    2) It is related to the previous comment but I think that the second section "Basic methods" could be enriched with some comments from the introduction.

    3) I think that the third section is interesting with intuitive proofs. However, I regret that there is no discussion on the stated theorems: is there any similar result in the literature, was it expected, is it tight?

    4) The fourth section seems to come bit out of nowhere and lacks a proper introduction. As a reader, it can seem odd to come from superlinear convegence rates (which is in the name of the paper) to adaptive techniques and a new method without a paragraph that bridges both sections. As said before, Theorem 4.1 should be commented and I would have expected a comparison with the results ensured by vanilla NATM.

    5) I believe that the fifth section should be merged with the fourth one or defined as a subsection.

**Questions:**

How do the theoretical results proved in the paper compare to the literature? Was this setting already studied?

* Minor comments/typos:
    - p.3, l.140: $c_3$ is never introduced before.
    - p.4, l.188,196,201: "the function $f(x)$" should be " the function $f$"
    - p.5, l.263: $e$ is both the vector of all ones and $1e-4=10^{-4}$
    - p.6, l.283: idem
    - p.6, l.287: the bigger $\rightarrow$ the larger
    - p.6, l.288: repetition "first, the first..."
    - p.6, l.293: I think the paragraph is a bit too long especially the explanation on gradient descent.
    - p.6, l.321: I understand but I think it is not very clear.
    - p.7, l.331: " has next constant" is a bit odd to me
    - p.7, l.338: "second Theorem" $\rightarrow$ "second theorem"
    - p.7, l.338: the sentence is too long and contains two times "hence"
    - p.18, l.930: "subsolover"
    - p.24, l.1274: "Optimal" seems to be a typo.

---

> ### Author Response · Authors · 2024-11-21
>
> Dear Reviewer BoHs,
>
> Thank you for your detailed and thoughtful review of our paper! We are pleased to hear that you found our theoretical analysis and practical developments compelling. We appreciate your constructive comments on the presentation style and organization of the paper, and we have made adjustments in the revised version to address them. For a detailed overview of the changes, please refer to the common response and revised version of the paper. Below, we specifically address your comments and the improvements we have made.
>
> ____
>
> ## Major Comments on Structure:
>
> ### Revised Introduction
> We have significantly shortened and reorganized the Introduction section, removing many of the equations that were initially included. These equations have been relocated to Section 2, as you suggested, for better high-level structure and readability.
>
> ### Expanded Methods Section
> The second section has been enriched with content previously found in the Introduction. We have added further details and descriptions, particularly regarding the OPTAMI library, to provide a more comprehensive and logically structured presentation of the methods.
>
> ### Novelty
> Our results, to the best of our knowledge, are novel and somewhat unexpected. Initially, we found it surprising that in the experiments, Cubic Regularized Newton methods exhibited superlinear convergence even when far from the solution. Upon further research, we identified a loophole in existing lower bounds that does not cover the convergence of second-order methods before reaching the quadratic convergence area.
> Building on this observation, we developed and presented a new convergence theory demonstrating global superlinear convergence. Additionally, we relaxed the strong convexity assumptions to the more general strongly star-convexity, which allows for non-convex cases. We have added Table 1 at the end of the manuscript to emphasize the novelty of our results compared to the current state of the literature. The lower bound for convergence of second-order methods for any precision is still an open question.
>
> ### Revised Fourth Section
>
> We reorganized Section 4, incorporating the recommended connections and enhancing clarity. Specifically, we added a comparison with vanilla NATM. For clarity, we note that the theoretical convergence rates of NATA and NATM are the same, differing only in the additional iterations required for the adaptive search of $\nu^t$ in NATA.
> Merged Sections and Additional Experiments:
> Following your recommendation, we merged Sections 4 and 5. Furthermore, we have introduced additional experiments on acceleration methods for regularized logistic regression (strongly convex case) in Figure 5. These experiments demonstrate global superlinear convergence for many methods, naturally leading to the theoretical proofs of superlinear convergence presented in the subsequent section.
>
> ## Minor comments/typos
> We also carefully reviewed the manuscript and corrected all the typos you   identified. Thank you for bringing them to our attention, as this helped us improve the overall quality of the paper.
>
> _____
>
> We hope these revisions address your concerns and improve the clarity and quality of the manuscript. Thank you again for your insightful comments, which have been instrumental in refining our work. Please do not hesitate to let us know if there are further areas requiring clarification or improvement.

---

> > ### Comment · Reviewer_BoHs · 2024-11-26
> >
> > I would like to thank the authors for these comments and their considerable efforts to rework the manuscript. I am very satisfied with the current state of the paper and I reconsidered my rating accordingly.

---

> > > ### Author Response · Authors · 2024-11-26
> > >
> > > Dear Reviewer BoHs,
> > >
> > > Thank you very much for your kind words and for reconsidering your rating. We greatly appreciate your thoughtful feedback and the opportunity to improve our paper.
> > >
> > > Best wishes, Authors.

---

### Author Response · Authors · 2024-11-21
**Modified Structure of The Work**

Dear Reviewers,

Thank you for your valuable feedback! A significant part of your comments concerns the structure of the paper and our approach to presenting the material. We greatly appreciate your helpful suggestions and prepared a new version of the article, taking your remarks into account. We hope this updated version clarifies the narrative behind the work and improves the overall structure.

## The Story Behind

We regret that the narrative we attempted to show in our initial submission was unclear. While we have significantly revised the introduction to make the story more comprehensible, we would like to outline it here for clarity.

The OPTAMI library was developed with the goal of unifying the implementation of second-order and higher-order optimization methods to enable consistent and fair comparisons. During its development, we encountered two open challenges:
Acceleration techniques often fail to improve practical performance, despite their theoretical appeal.
For strongly convex problems (e.g., regularized logistic regression), methods exhibit global superlinear convergence in practice, which contradicts theoretical upper bounds and appears to fall outside the scope of existing lower bounds.
These challenges motivated us to develop practical and theoretical solutions, which we present alongside the new library.

## Reorganization

The new organization of the paper is as follows:
### Introduction
The introduction has been shortened by removing technical details and divided into clear paragraphs. We also created subsection about practical questions, which includes the introduction of OPTAMI library and open challenges we faced developing the library.
### Methods and Notation
This section has been expanded to include some of the technical details previously found in the introduction. It also provides a detailed description of the OPTAMI library.
### Improving Practical Performance of Accelerated Methods
This section consolidates Sections 4 and 5 from the initially submitted version. It begins with a discussion of the algorithms implemented in the OPTAMI library and addresses practical limitations of existing acceleration schemes, starting with Nesterov acceleration. We then introduce a novel algorithm, NATA, specifically designed for improved practical performance, and prove its convergence. Finally, we present an experimental comparison of five different acceleration techniques.
### Global Superlinear Convergence of High-Order Methods for Strongly Star-Convex Functions
Strongly convex experiments discussed in the previous section demonstrate superlinear convergence. In this section, we present a new theoretical result that explains this behavior.
### Conclusion

---

### Meta-Review · Area_Chair_a1BK · 2024-12-19

**Metareview:**

The paper focuses on the analysis of high order methods for mu-strongly star-convex functions. They propose an adaptive variant of Nesterov Accelerated Tensor Method called NATA. The authors also provide OPTAMI, a python library for high order methods. The reviewers found the paper to be solid, though with some concerns that the presentation of the paper is trying to cover too much ground. Based on my own reading, the contribution of the paper is solid and worthy of acceptance.

**Additional Comments On Reviewer Discussion:**

Most concerns were relating to the presentation, which I believe have been adequately addressed.

---

### Decision · Program_Chairs · 2025-01-22

Accept (Poster)